# Multimodal Web Navigation with Instruction-Finetuned Foundation Models

**Hiroki Furuta**[1,2*] **Kuang-Huei Lee**[2] **Ofir Nachum**[2] **Yutaka Matsuo**[1]
**Aleksandra Faust**[2] **Shixiang Shane Gu**[1,2] **Izzeddin Gur**[2]
[1]The University of Tokyo  [2]Google DeepMind
furuta@weblab.t.u-tokyo.ac.jp

## Abstract

The progress of *autonomous web navigation* has been hindered by the dependence on billions of exploratory interactions via online reinforcement learning, and domain-specific model designs that make it difficult to leverage generalization from rich out-of-domain data. In this work, we study data-driven offline training for web agents with vision-language foundation models. We propose an instruction-following multimodal agent, WebGUM, that observes both webpage screenshots and HTML pages and outputs web navigation actions, such as *click* and *type*. WebGUM is trained by jointly finetuning an instruction-finetuned language model and a vision encoder with temporal and local perception on a large corpus of demonstrations. We empirically demonstrate this recipe improves the agent's ability of grounded multimodal perception, HTML comprehension, and multi-step reasoning, outperforming prior works by a significant margin. On the MiniWoB, we improve over the previous best offline methods by more than 45.8%, even outperforming online-finetuned SoTA, humans, and GPT-4-based agent. On the WebShop benchmark, our 3-billion-parameter model achieves superior performance to the existing SoTA, PaLM-540B. Furthermore, WebGUM exhibits strong positive transfer to the real-world planning tasks on the Mind2Web. We also collect 347K high-quality demonstrations using our trained models, 38 times larger than prior work, and make them available to promote future research in this direction.

## 1 Introduction

Web navigation is a class of sequential decision making problems where agents interact with web interfaces following user instructions (Shi et al., 2017; Liu et al., 2018; Gur et al., 2019). Common web navigation tasks include, for example, form filling (Diaz et al., 2013), information retrieval (Nogueira & Cho, 2016; Adolphs et al., 2022), or sending emails via a sequence of interactions with computer interface such as *click* or *type* (Figure 1). Recently, there has been a growing interest in developing agents to automate these actions and free humans from repetitive interactions (Mazumder & Riva, 2020; Li et al., 2020; Shvo et al., 2021).

Most prior works studied web navigation problems as online RL to learn the optimal action distribution with task-specific models from scratch (Liu et al., 2018; Gur et al., 2019; Jia et al., 2019; Humphreys et al., 2022). However, online RL requires massive trials-and-errors and is often infeasible in practice since the failure in web navigation would result in undesirable consequences; for instance, wrong password may lead to account freeze, and sending email to the wrong person could be problematic in a business scene. In contrast, offline training from the static dataset enables safe development of web agents, but the performance has been sub-optimal compared to those online RL counterparts (Humphreys et al., 2022; Gur et al., 2022). Furthermore, many of the prior works was unable to leverage rich out-of-domain data for generalization, as they usually use specialized models to explicitly handle the hierarchical structures of document object model (DOM) and their dependencies, for example, with LSTM (Gur et al., 2019; 2021), self-attention (Liu et al., 2018), or GNN (Jia et al., 2019). And many of them only output a fixed set of categorical actions (Humphreys et al., 2022), which is unfavorable for truly open-ended web navigation in the real world.

---

*Work done as Student Researcher at Google.

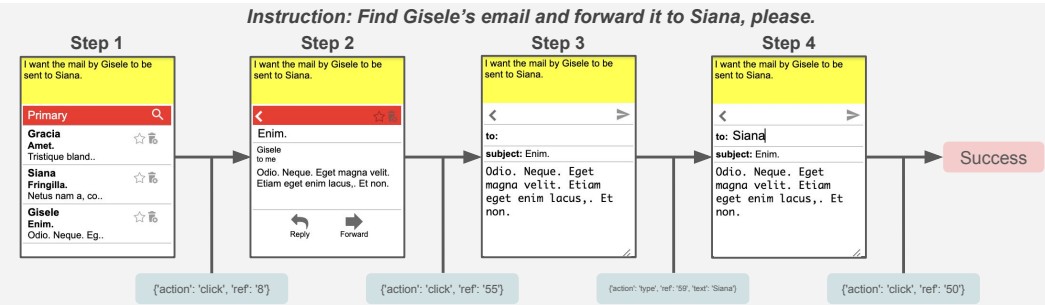

Figure 1: Example episode on MiniWoB++ (Shi et al., 2017; Liu et al., 2018) (`email-inbox-forward-nl`). The agent clicks the email from the proper sender, and types the correct receiver to forward that email, to satisfy the given instruction (e.g. *Find Gisele's email and forward it to Siana, please*). WebGUM makes use of both HTML and image screenshot information to adapt a pre-trained instruction-finetuned foundation model to solve challenging web-based tasks.

Recently, foundation models (Bommasani et al., 2021), especially large language models (LLM) (Brown et al., 2020; Chowdhery et al., 2022), have demonstrated superior performance in commonsense, symbolic, arithmetic, and multi-step logical reasoning (Wei et al., 2022b;c; Kojima et al., 2022). These models enable transformative generalization and are capable of solving wide ranges of interactive decision making problems in the wild, including but not limited to task planning in robotics (Huang et al., 2022a;b; Shah et al., 2022; Ahn et al., 2022), board game (Meta Fundamental AI Research Diplomacy Team et al., 2022), web-based retrieval and browser crawling (Nakano et al., 2021; Yao et al., 2022b; Zaheer et al., 2022).

In this work, we leverage pre-trained vision and language foundation models and introduce a competitive offline learning recipe for autonomous web agents: First, we hypothesize that grounded spatial understanding is important for web navigation (Humphreys et al., 2022; Toyama et al., 2021) and thus enables our agent to observe both HTML and screenshots by combining a language model and a ViT (Dosovitskiy et al., 2020), from semantically rich multimodal tokens that perceive local and temporal information. Second, we observe that web navigation tasks are by nature instruction-following and thus base the language model on an instruction-tuned LLM (Wei et al., 2022a; Chung et al., 2022; Ouyang et al., 2022; Iyer et al., 2022) instead of self-supervisedly pre-trained LLMs (Raffel et al., 2020; Brown et al., 2020) as in Gur et al. (2022). Third, we collect a large multimodal corpus, with both HTML and screenshots, to finetune the language model and ViT jointly. Fourth, our model outputs action in free-form text. These four key pieces together give us a multimodal web agent, which we call *Web navigation via Grounded Understanding Models* or WebGUM in short. As shown in Figure 1, our model takes in a command for a web-based task via a natural language instruction (e.g., in an email client, *Find Gisele's email and forward it to Siana, please.*) and uses multimodal observations of the computer interface to complete the task via a sequence of computer actions.

On MiniWoB++ (Shi et al., 2017; Liu et al., 2018), a simulated web navigation environment benchmark, WebGUM outperforms previous best offline approaches trained with HTML inputs (Gur et al., 2022) by 45.8%, and even the best existing online RL approaches (Humphreys et al., 2022), despite being trained fully offline with much fewer experiences. WebGUM also shows better performance than humans and private-LLM-based agents (Kim et al., 2023; Sun et al., 2023). We perform extensive ablations and analysis in Section 5 to demonstrate WebGUM's advantages in (1) **temporal and local multimodal perception**, (2) **dataset and model size scaling**, (3) **better HTML understanding**, and (4) **ability of multi-step reasoning**. WebGUM grounds vision and HTML understanding on the computer interface, which is critical for solving multi-step tasks with dynamic page transitions or tasks that require visual contexts, such as booking flights (+50%), shape recognition (+22%), or crawling social media (+21%). Using instruction-finetuned language models (Chung et al., 2022), compared to using vanilla models (Raffel et al., 2020), improves the success rate on MiniWoB++ by 25%, and is especially adept at handling the unknown composition of the tasks or out-of-distribution HTML inputs synthesized with realistic perturbations. On the WebShop benchmark (Yao et al., 2022a), we demonstrate that the capability of multi-step reasoning (Wei et al., 2022c) in language models enables better performance than existing state-of-the-art few-shot PaLM-540B (Yao et al., 2022b; Chowdhery et al., 2022), while our model only has 3 billion parameters. WebGUM exhibits strong positive transfer to the real-world action prediction tasks on the Mind2Web while surpassing

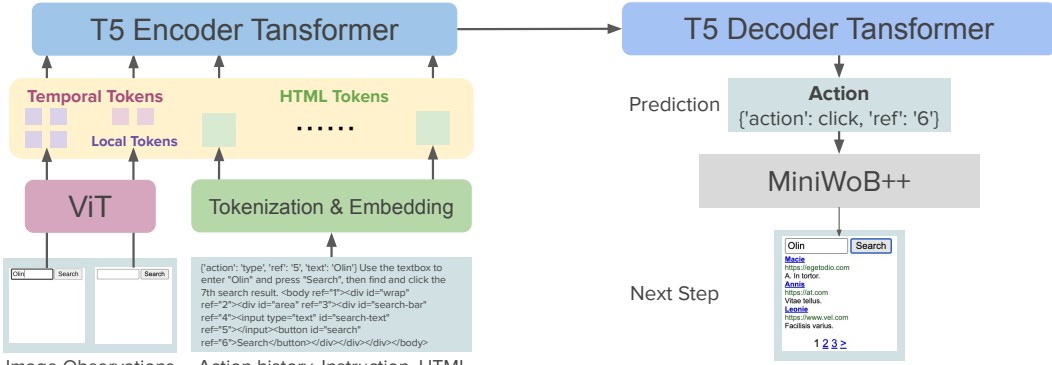

Figure 2: Overview of WebGUM, our multimodal encoder-decoder model. It takes screenshots, action history, instruction, and HTML as inputs. Image observations are embedded to tokens via pre-trained vision transformer (ViT) (Dosovitskiy et al., 2020). Visual tokens contain rich temporal information from recent $H$-step ($H = 2$) and local information from $16 \times 16$-size patches. Multimodal language-image tokens are fed into pre-trained T5 encoder-decoder models (Raffel et al., 2020), and are jointly trained to predict executable actions in text formats.

GPT-4. Finally, we collect 347K multimodal expert demonstrations on MiniWoB++, 38 times larger than the existing unimodal dataset (Liu et al., 2018), and make these publicly available for future research [1]. We believe that incorporating foundation models for efficient offline training is a scalable approach towards real-world web automation where online interactions are prohibitively costly.

## 2 RELATED WORK

**Web Navigation** Among many proposed benchmarks for autonomous web navigation (Toyama et al., 2021; Burns et al., 2022; Yao et al., 2022a), one of the most inclusive and representative benchmark to test the capability of autonomous agents is MiniWoB++ (Shi et al., 2017; Liu et al., 2018), which consists of a set of simulated websites with various user instructions from primitive tasks to complex multi-step decision making tasks, such as sending emails or booking flights. Prior works have tried to solve this benchmark using a variety of techniques; Liu et al. (2018) and Gur et al. (2019; 2021) leverage the guidance during online RL from high-level workflow (Liu et al., 2018) or curriculum learning (Gur et al., 2019; 2021), which should be, however, designed per task, and then would not be scalable methods. Other approaches have employed supervised learning (SL) with a large million-scale dataset and following RL-finetuning (Humphreys et al., 2022), or SL with LLM-based agents (Gur et al., 2022). Offline SL agents often suffer from sub-optimal behavior, and online RL with tremendous exploratory experiences has been critical for proficient navigation on the web (Humphreys et al., 2022), which is, however, difficult to conduct in real websites as there is typically no reward signal and interactions are prohibitively costly. As shown in Appendix I, many of these approaches depend on task-specific hierarchical structures of DOM (Jia et al., 2019; He et al., 2020), tailored architectures to encode their dependencies such as LSTM (Gur et al., 2019; 2021), self-attention (Liu et al., 2018), or GNN (Jia et al., 2019), and task-dependent categorical output space (Humphreys et al., 2022), which could not handle open-ended multi-task settings similar to real world, or incorporate pre-trained models. In contrast, we remove such web-specific architectures and convert web navigation into visual question-answering format (text, image → text), which allows us to leverage pre-trained foundation models (Chung et al., 2022; Dosovitskiy et al., 2020) as rich prior knowledge on the web, and then to learn the capable agents even with offline training.

**Large Language Models for Web Navigation** Concurrently, private-LLM-based agents, such as InstructGPT (text-davinci-003) (Ouyang et al., 2022) and GPT-3.5-turbo, have achieved competitive performance to RL-fintuned models and humans by leveraging a handful of few-shot demonstrations with self-improvement (Kim et al., 2023), code generation (Sun et al., 2023), and structured prompts (Zheng et al., 2023). In contrast, WebGUM focuses on multimodality and finetuning with domain-specific data. With those, we show very competitive performance compared to PaLM-540B with only 3 billion parameters. WebGUM can also handle long HTML observation tasks, such as `book-flight` or `choose-date-hard`, where agents that rely on in-context few-shot learning tend to run out of input tokens. In addition, our models do not requires ad-hoc prompt engineering.

---

[1]https://console.cloud.google.com/storage/browser/gresearch/webllm

| Methods | Modality | Pre-trained Models | Offline | Dataset | Success Rate |
|---------|----------|--------------------|---------|---------|--------------|
| CC-Net (SL) | DOM+Image | ResNet | ✔ | 2400K | 32.0% |
| WebN-T5 | HTML | T5-XL | ✔ | 12K | 48.4% |
| WebGUM (Ours) | HTML+Image | Flan-T5-Base,ViT-B16 | ✔ | 2.8K | 61.1% |
| | HTML | Flan-T5-XL | ✔ | 401K | 88.7% |
| | HTML+Image | Flan-T5-XL,ViT-B16 | ✔ | 401K | **94.2%** |
| WGE | DOM | – | ✗ | 12K+ | 64.6% |
| CC-Net (SL+RL) | DOM+Image | ResNet | ✗ | 2400K+ | 93.5% |
| Human | – | – | – | – | 93.5% |
| RCI | HTML | GPT-3.5-turbo | ICL | ∼0.1K | 90.6% |
| AdaPlanner | HTML | text-davinci-003 | ICL | ∼0.1K | 92.9% |
| RCI | HTML | GPT-4 | ICL | ∼0.1K | 94.0% |
| Synapse | HTML | GPT-3.5-turbo | ICL | ∼0.1K | **98.5%** |

Table 1: Average success rate on MiniWoB++. We refer to Zheng et al. (2023) for the baseline performances. See Appendix G for the detailed scores. WebGUM outperforms the previous finetuned-LLM with 3B parameters (Gur et al., 2022), which is the best among offline methods, even with 2.8K dataset and Base-size model (310M parameters). Scaling dataset and model size, WebGUM beats the online RL-finetuned state-of-the-art (Humphreys et al., 2022) despite fully offline training, and exceeds humans or LLM-based agents with GPT-4 (Kim et al., 2023). "+" in Dataset column means extra billions of frames are required during the online RL phase.

In Appendix B, We discuss additional related works on multimodal large-scale models and foundation models for decision making.

## 3 PRELIMINARIES

We formulate autonomous web navigation as a deterministic sequential decision making problem; composed of a state space $\mathcal{S}$, action space $\mathcal{A}$, deterministic transition function $T : \mathcal{S} \times \mathcal{A} \to \mathcal{S}$, instruction space $\mathcal{G}$, reward function (or episodic success criteria) $r : \mathcal{S} \times \mathcal{G} \times \mathcal{A} \to \{0, 1\}$. At each time step $t$, the agent follows a parameterized policy conditioned on previous states and actions $\pi : \underbrace{\mathcal{S} \times \cdots \times \mathcal{S}}_{\times t} \times \underbrace{\mathcal{A} \times \cdots \times \mathcal{A}}_{\times t} \times \mathcal{G} \to \mathcal{A}$, and transits to the next state: $s_{t+1} = T(s_t, a_t)$. This process continues until the agent reaches the terminal state (e.g. Submit button is clicked) or the max time step is exceeded. An episode is treated as a success if given instruction $g$ is satisfied (i.e. $r(s_t, g, a_t) = 1$), and as a failure if the agent takes a invalid action or reaches a wrong terminal state.

In autonomous web navigation, the state $s_t \in \mathcal{S}$ is a web page consisting of the raw HTML as a text sequence and a screenshot as an image. Following prior works (Shi et al., 2017; Liu et al., 2018; Gur et al., 2019; 2021), we assume the constraint action space: function(selector, text). function is either *click* or *type*, selector is an integer index that can uniquely specify the element, and text is a text input for *type* function.

Figure 1 presents an example episode of MiniWoB (Shi et al., 2017), which involves multi-step decision making. To meet the given instruction, the agent clicks an email from the proper sender and types the correct receiver to forward that email. MiniWoB also has primitive behavioral tasks such as clicking buttons or entering texts. For the examples of WebShop (Yao et al., 2022a), see Appendix L.

## 4 WEBGUM

### 4.1 MULTIMODAL TRANSFORMER MODELS WITH TEMPORAL AND LOCAL PERCEPTION

In this work, we follow Gur et al. (2022) to use T5 (Raffel et al., 2020), an encoder-decoder architecture, for HTML-based web navigation, as its bi-directional nature could be a good fit for the tree structure of HTML and the architecture has been shown to scale well. We combine T5 with a vision transformer (ViT) (Dosovitskiy et al., 2020) for multimodality as illustrated in Figure 2. Specifically, we use the ViT to map image observations (screenshots) into image tokens. The ViT is pre-trained on ImageNet-21K classification (Deng et al., 2009). The T5 encoder then consumes both visual and HTML tokens in a unified manner, and the decoder predicts actions in text. See Appendix C for more implementation details.

| Methods          | Modality   | Success Rate |
| ---------------- | ---------- | ------------ |
| WebGUM          | HTML       | 88.7%        |
| WebGUM (white)   | HTML+Image | 90.9%        |
| WebGUM (random)  | HTML+Image | 92.2%        |
| WebGUM          | HTML+Image | 94.2%        |

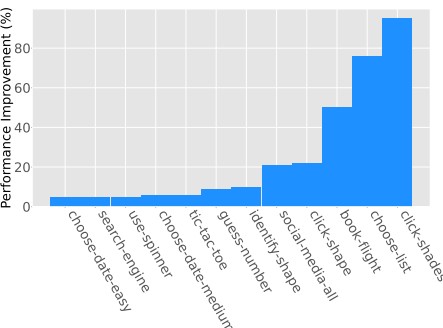

Figure 3: **(Left)** Average success rate with white/random image inputs. The results imply that WebGUM successfully leverages multimodal information from temporal and local perception tokens. **(Right)** Top-10 performance improvement among MiniWoB++ by adding image modality to HTML. We subtract the success rates to compute absolute improvement: `(SR of WebGUM(HTML+Image)) - (SR of WebGUM(HTML))`. Image modality is leveraged for multi-step tasks with dynamic page transitions or tasks that require visual concept understanding (e.g. `book-flight` or `click-shape`). See Appendix G and L for the details.

**Encoding Temporal and Local Visual Tokens** For language models to be aware of task temporal information and local scene recognition, the encoder considers multimodal tokens extracted from a history of patched screenshots ($H = 2$ steps). Temporal visual tokens contribute to predict the consistent actions in a multi-step tasks. To better extract spatial and semantic information across the local parts of websites, our ViT encodes one local token per patch rather than global one per image (i.e. CLS-token). We divide an input image into $16 \times 16$ patches – giving a total of $14 \times 14$ (number of patches) $\times 2$ (temporal window) $= 392$ visual tokens. We crop the screenshots of MiniWoB++ to remove the yellow instruction part, and the image size becomes $160 \times 160$. We pad cropped images with white pixels to fit them into $224 \times 224$; the default input size for ViT.

## 4.2 INSTRUCTION-FINETUNED LARGE LANGUAGE MODELS

We base our language model on Flan-T5 (Chung et al., 2022), an instruction-finetuned T5, as opposed to using a vanilla pre-trained T5 as in Gur et al. (2022). Flan-T5 is finetuned with large-scale instruction-following format problems and chain-of-thought examples across a variety of domains, including reasoning or programming. Considering that web navigation is inherently an instruction-following task, we hypothesize that carefully trained instruction-finetuned models could generalize well to enhance the alignment with user instruction and zero-shot reasoning in the web-navigation, interactive decision making context. For the same reason, we also hypothesize that these high-performing instruction-finetuned models enable better sample efficiency and downstream performance, and thus are well-suited for offline learning. We further finetune the Flan-T5 language model and the ViT vision encoder jointly (Figure 2) on a large corpus of instruction-following multimodal web navigation data, which we describe in Section 4.3. In Section 5, we empirically demonstrate that this instruction-finetuned recipe improves HTML comprehension, multi-step reasoning and decision making significantly.

## 4.3 LARGE-SCALE DATA COLLECTION WITH LANGUAGE MODEL AGENTS

Recent successes of foundation models are largely powered by internet-scale data (Brown et al., 2020; Radford et al., 2021; Chen et al., 2022; Wang et al., 2023). While large amount of data is critical, for web navigation domain, there is only a small public dataset for MiniWoB++, consisting of 12K episodes of human demonstration (Liu et al., 2018). Moreover, the dataset only consists of DOM observations and lacks any visual features, which might limit the fine spatial perception of the elements on the page. A large-scale multimodal dataset, including screenshots of websites, is required to build a better navigation policy at scale.

To collect a huge amount of multimodal behavioral dataset on MiniWoB++, we leverage the finetuned-LLM policy from Gur et al. (2022), instead of human demonstrators (Liu et al., 2018; Humphreys et al., 2022). This significantly reduces the cost to construct a new dataset by leveraging the prior success of autonomous agents. We first rollout a LLM policy with 100 episodes per task, which results in a 2.8K successful episodes. Then, we finetune Flan-T5-XL models with this small dataset and run those with 10,000 episodes per task. Lastly, we collect additional 54K demonstrations

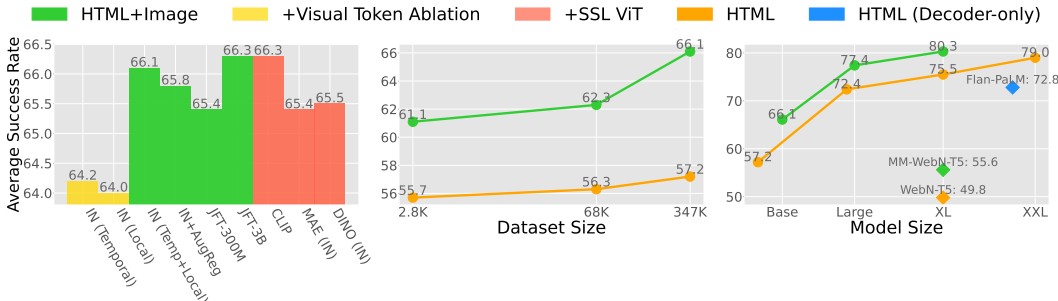

Figure 4: Average success rate of WebGUM with visual perception tokens and ViT pre-training ablations (left), different dataset size (middle) and model architectures (right). In dataset and model size results, X-axis is a logarithmic scale. **(left)** While the effects of various pre-trained ViT with different datasets or self-supervised objectives are marginal, employing both temporal and local perception tokens is critical for the performance. **(middle & right)** As for both HTML and multimodal models, we could observe the scaling effect: the larger the dataset and model size are, the higher the success rates are. The results also prove that decoder-only Flan-PaLM-8B is not as good as similar-size encoder-decoder models.

with Synapse (Zheng et al., 2023), a private-LLM-based agents with prompting, for the tasks where the finetuned-LLM may not complete well. Such efforts result in a multi-task dataset with 401K (347+54K) episodes including HTML and screenshots at each step. See Appendix F for more details.

## 5 RESULTS

We test our method on MiniWoB++ (Shi et al., 2017; Liu et al., 2018) with 100 evaluation episodes per task, taking the average success rate over 56 tasks taken from Gur et al. (2022). Table 1 shows that WebGUM, with a small 2.8K dataset and Base-size model (310M parameters), significantly outperforms previous offline methods for web navigation (Humphreys et al., 2022; Gur et al., 2022). While they used 2.4 million episodes or 3 billion parameters, WebGUM could improve the data and parameter efficiency to achieve superior performance in offline regime, which is realized by the problem simplification of web navigation in order to leverage temporal-local visual perception and instruction-finetuned LLMs as strong inductive bias on web environments. In addition, scaling dataset and model size, WebGUM achieves 94.2% success rate[2], exceeding the previous best offline model, WebN-T5 (Gur et al., 2022), by over 45.8% and even surpassing the online RL-finetuned SoTA, CC-Net (Humphreys et al., 2022) (+0.7%), despite our fully offline training and much fewer data. Moreover, WebGUM surpasses humans and recent LLM-based agents, such as RCI (Kim et al., 2023) and AdaPlanner (Sun et al., 2023), even with GPT-4 (OpenAI, 2023). The per-task comparison and error analysis (Appendix G, L) imply that there is room for improvement in complex reasoning tasks requiring memory such as guess-number.

In the following sections, we perform extensive and precise ablations of WebGUM to clearly identify the source of improvement. Especially, we will demonstrate the contribution of (1) **temporal and local multimodal perception** (Section 5.1), **architectures and pre-trained models**, and (2) **dataset and model size scaling** (Section 5.2). We will also point out (3) **better HTML comprehension** (Section 5.3) and (4) **capability of multi-step reasoning** (Section 5.4) from instruction-finetuned LLMs. Furthermore, we prove that WebGUM can be transferable to the real-world tasks (Section 5.5).

### 5.1 TEMPORAL AND LOCAL VISUAL PERCEPTION FOR GROUNDED WEB NAVIGATION

To verify the importance of image modality, we design three ablations: (i) input replacement, (ii) removing visual perception tokens, and (iii) employing different pre-trained ViT. We first replace image observations with completely white images, and with randomly sampled MiniWoB++ screenshots taken in the initial states at test time. For visual token and pre-trained ViT ablations, we prepare various pre-trained weights with ImageNet-21K (IN) + AugReg (Steiner et al., 2022), JFT-300M (Sun et al., 2017), or JFT-3B (Zhai et al., 2022), and with self-supervised objectives such as CLIP (Radford et al., 2021), MAE (He et al., 2021), or DINO (Caron et al., 2021), and then finetune Base-size models as a proxy of larger-size models (Hoffmann et al., 2022) to reduce the computational costs.

---

[2]Videos are available at https://sites.google.com/view/mm-webnav/

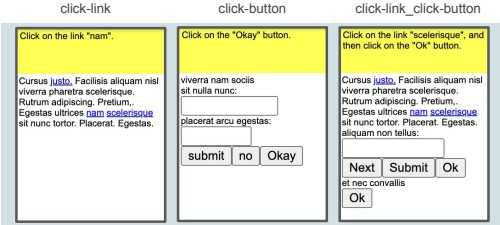

| Methods | Modality | Success Rate |
|---|---|---|
| WebN-T5 (Gur et al., 2022) | HTML | 51.0% |
| Synapse (Zheng et al., 2023) | HTML | 73.8% |
| WebGUM | HTML | 74.2% |
| WebGUM | HTML+Image | **78.5%** |

Figure 5: **(Left)** Example of compositional evaluation on MiniWoB++. We combine two different tasks (`click-link` and `click-button`) into a single sequential task (`click-link_click-button`) at test time (see Appendix H). **(Right)** Average success rate on 6 compositional MiniWoB tasks. WebGUM generalizes combinational tasks better than Gur et al. (2022) and Zheng et al. (2023), a SoTA LLM-agent in MiniWoB++.

| Methods | Modality | Perturbation | Success Rate |
|---|---|---|---|
| WebN-T5 (Gur et al., 2022) | HTML | Top | 24.7% |
| | | Bottom | 42.8% |
| | | Coordinates | 6.4% |
| WebGUM | HTML | Top | 53.6% |
| | | Bottom | 48.0% |
| | | Coordinates | 39.8% |
| WebGUM | HTML+Image | Top | **71.8%** |
| | | Bottom | **64.7%** |
| | | Coordinates | **62.6%** |

Figure 6: **(Left)** Example of input perturbation for MiniWoB++ evaluation, taken from `click-button`. We prepare three different types of perturbations at test time: adding extra HTML at the top of the original input HTML (left) or at the bottom (middle), and adding task-irrelevant attributes in each element (right) such as coordinate information (left, right, top, bottom). **(Right)** Average success rate of perturbation evaluation on MiniWoB++. The results reveal that while all the methods are affected by input corruptions to some extent, WebGUM, especially with multimodality, achieves significantly better performances than previous method.

In Figure 3 (left), the performance of the model with white images is comparable to the unimodal model. Presumably because the model with randomly-taken images may accidentally contain the images from the target task, WebGUM (random) slightly surpasses WebGUM (white). These results prove WebGUM successfully obtains grounded vision and HTML understanding by leveraging temporal and local fine perception. In the visual token ablation, Figure 4 (left) shows that combining both temporal and local visual tokens (66.1%) improves the performance than temporal (64.2%) or local tokens only (64.0%). Interestingly, the effects of different pre-trained ViT are marginal, compared to visual tokens, which highlights our contribution on designing suitable architecture for multimodal web navigation.

We also compare per-task performance gaps caused by adding vision modality to language models. Figure 3 (right) presents top-10 absolute performance improvement, suggesting WebGUM leverages visual inputs for multi-step tasks with dynamic page transitions (e.g. `book-flight`; +50%) or tasks requiring visual context understanding (e.g. `click-shape`; +22%) (see Appendix G and L).

## 5.2 SCALING EFFECT IN DATASET AND MODEL SIZE

In this section, we show the importance of scaling up the dataset and model size in WebGUM, similar to the observations in the language and vision domain (Shoeybi et al., 2019; Kaplan et al., 2020; Rae et al., 2021; Wei et al., 2022b; Chowdhery et al., 2022). To investigate data scaling, we prepare three dataset: minimal 2.8K demonstrations, 347K demonstrations, and its 20%-size demonstrations (68K), and then finetune Flan-T5-Base with them. Figure 4 (middle) proves that increasing dataset size leads to the improvement of success rate. Because multimodal models benefit from the scaling more, the larger dataset size might be more crucial in multimodal models, which also supports our attempts to construct large-scale multimodal dataset for web navigation. Notably, Base-size WebGUM with 2.8K episodes already achieves 55.7%/66.1%, surpassing previous best SL models (49.8%/55.6% we trained with 347K episodes). This surprising data efficiency comes from the sufficient inductive bias and alignment with the user intentions in instruction-finetuned LLMs.

In addition to dataset size, Figure 4 (right) shows that the performance of WebGUM improves as the number of parameters in T5 model increases from Base (220M) to XXL (11B). These results also reveal that scaling the models might be more important than the dataset; the low-capacity model may cap the performance at a lower level. In contrast, decoder-only Flan-PaLM-8B only

achieves 72.8% success, comparable to WebGUM-Large (770M), which emphasizes the advantage of encoder-decoder models in web navigation. See Appendix D for further details.

## 5.3 Better HTML Comprehension from Instruction-Finetuned LLMs

We have demonstrated that instruction-finetuned LLMs outperforms vanilla LLMs in web navigation. To analyze the effect of instruction-finetuning more precisely, we here focus on the capability of HTML understanding. Since instruction-finetuned LLMs perform well on many NLP tasks with content comprehension (Chung et al., 2022; Iyer et al., 2022), web navigation should also benefit from them. As a test bed for HTML comprehension, we investigate (1) generalization to unseen compositions of known tasks, and (2) robustness to the realistic input perturbations, which are also important challenges for the web agents to be deployed on the real-world internet. We also provide the base language model comparison on a standard HTML comprehension benchmark, WebSRC (Chen et al., 2021d) in Appendix E, where Flan-T5 achieves better EM/F1 scores than T5 after finetuning.

For the compositional tasks, we pick up 4 `click`-"something" (link, button, checkboxes, dialog) tasks and make 6 combinations of these by naively stitching with 2 or 3 tasks (e.g. Figure 5). See Appendix H for further details. The results show that WebGUM with HTML and image inputs outperforms prior finetuned-LLM (Gur et al., 2022) and Synapse (Zheng et al., 2023), a SoTA LLM agent in MiniWoB++, which implies WebGUM has obtained better reading skills for web navigation and could transfer them to handle unseen HTML in compositional tasks robustly.

To test the robustness against input corruptions, we test three different realistic perturbations; adding extra HTML at the top or bottom of the original HTML, and adding attributes of coordinates (left, right, top, bottom; they are unrelated to solving the tasks) in each element of HTML at test time. These perturbations often happen in the real world due to the renewal or API changes, not to mention unknown websites, but rule-based pre-processing may not fully cover them. The results show that while all the methods are affected by the input corruptions to some extent, WebGUM, with both HTML and HTML plus image modalities, achieves significantly better performances than Gur et al. (2022). Notably, WebGUM outperforms prior finetuned LLM (+ 56.2% in multimodal and +33.4% in unimodal models) even when extra distracted attributes are added to HTML. They support our hypothesis: instruction-finetuning imporves HTML comprehension in LLMs, which enables the downstream agents to deal with out-of-distribution inputs or tasks robustly.

## 5.4 Ability of Multi-Step Reasoning as a Prior for Interactive Decision Making

Another notable feature in instruction-finetuned LLMs is an ability of multi-step reasoning (Chung et al., 2022). We hypothesize this reasoning capability would play an important role as a prior for interactive decision making. To decouple the evaluation of reasoning capability from visual page perception, HTML understanding, and the benchmark simulator (MiniWoB++), we extensively evaluate our WebGUM on WebShop (Yao et al., 2022a), another online-shopping website simulator with a large amount of real-world product data. Because it requires complex multi-step decisions considering previous contexts for item comparison, WebShop is suitable for investigating the capability of multi-step reasoning from instruction-finetuned LLM in depth (Yao et al., 2022a;b). WebShop provides a user instruction that

| Methods | Training | Models | Score | Success Rate |
|---------|----------|--------|-------|--------------|
| Rule | – | – | 45.6 | 9.6% |
| IL | SL | BART, BERT | 59.9 | 29.1% |
| IL+RL | SL+RL | BART, BERT | 62.4 | 28.7% |
| Act | In-context | PaLM-540B | 62.3 | 30.1% |
| ReAct | In-context | PaLM-540B | 66.6 | 40.0% |
| WebN-T5 | SL | T5-XL | 61.0 | 29.8% |
| WebGUM | SL | Flan-T5-XL | **67.5** | **45.0%** |

Table 2: Average score and success rate on Web-Shop (Yao et al., 2022a). WebGUM achieves 45.0% success, outperforming baseline methods including ReAct, a prompted PaLM-540B. We refer Yao et al. (2022b) for the baselines.

describes the features of item (e.g. *I need a long clip-in hair extension which is natural looking, and price lower than 20.00 dollars*). The agents should search, compare and choose a proper product that matches the given instruction. The performance score is evaluated by the percentage of required attributes covered by the chosen product, and if the product meets all the requirements, that episode is labeled a success. See Appendix K for further details.

Table 2 shows that WebGUM achieves 45.0% success, significantly outperforming not only simple baselines, such as supervised imitation learning (IL), IL plus RL-finetuing and WebN-T5 (by more than 15%), but also recent prompt-based LLM agents, including ReAct (Yao et al., 2022b) (i.e. PaLM-540B (Chowdhery et al., 2022) with one-shot prompt and reasoning annotations), while our

| | | Cross-Task | | | | Cross-Website | | | | Cross-Domain | | | |
|---|---|---|---|---|---|---|---|---|---|---|---|---|---|
| | Train | Ele. Acc | Op. F1 | Step SR | SR | Ele. Acc | Op. F1 | Step SR | SR | Ele. Acc | Op. F1 | Step SR | SR |
| GPT-4 | ICL | 41.6 | 60.6 | 36.2 | 2.0 | 35.8 | 51.1 | 30.1 | 2.0 | 37.1 | 46.5 | 26.4 | 2.0 |
| MindAct-Large | SL | 53.4 | 75.7 | 50.3 | 7.1 | 39.2 | 67.1 | 35.3 | 1.1 | 39.7 | 67.2 | 37.3 | 2.7 |
| MindAct-XL | SL | 55.1 | 75.7 | 52.0 | 5.2 | 42.0 | 65.2 | 38.9 | 5.1 | 42.1 | 66.5 | 39.6 | 2.9 |
| WebGUM-Large (ours) | SL | 55.3 | 78.9 | 51.9 | 7.5 | 43.6 | 70.3 | 39.3 | 5.1 | 42.8 | 70.6 | 40.2 | 2.9 |
| WebGUM-XL (ours) | SL | **57.2** | **80.3** | **53.7** | **8.5** | **45.3** | **70.9** | **41.6** | **5.2** | **43.9** | **72.2** | **41.4** | **3.2** |

Table 3: Action prediction evaluation in real-world Mind2Web dataset. We adopt the top-50 candidate generation results and direct QA formulation by following Deng et al. (2023). WebGUM, transferred from MiniWoB, demonstrates superior performance to MindAct and GPT-4 across task/website/domain generalization.

model only has 3 billion parameters. Due to the consistent reasoning and enhanced alignment with user intentions, WebGUM could compare the products with backtracking, and choose proper options (see Appendix L). Our results imply that ability of multi-step reasoning in Flan-T5 works as strong and transferable prior knowledge for downstream decision making.

## 5.5 STRONG TRANSFER TO REAL-WORLD ACTION PREDICTION

Lastly, we demonstrate the applicability of WebGUM to real-world problems. We test WebGUM on Mind2Web (Deng et al., 2023), a real-world demonstration dataset with about 2K instructions on 137 websites. In the action prediction tasks, we transfer WebGUM finetuned for MiniWoB++ with 401K dataset into real-world Mind2Web by further finetuning with the training set. WebGUM takes top-50 relevant HTML snippet candidates, instructions, and action history as inputs and outputs next actions by predicting the element id, operations (e.g. *click*, *type*), and values. Table 3 reveals that WebGUM, transferred from MiniWoB, achieves superior performance to MindAct-Large/XL and even GPT-4 in all the categories (cross-task/website/domain). Because both MindAct and WebGUM are based on Flan-T5, these results support that WebGUM exhibits strong positive transfer to real-world tasks.

## 6 DISCUSSION AND LIMITATION

Throughout the paper, we present an effective and practical methodology to simplify web navigation into offline training in order to leverage the inductive bias of web environments in instruction-finetuned LLMs. While WebGUM exhibits positive transferability to real-world problems in Mind2Web, we leave it as future work to scale multimodal foundation models into the deployment for real-world web navigation (Gur et al., 2023).

We collect and release a multimodal expert dataset with 347K episodes on MiniWoB++. However, this is still far from internet-scale dataset that is necessary for generalist models. Collecting behavioral data at scale by iterative data-collection and deployment (Ghosh et al., 2021; Matsushima et al., 2021; Li et al., 2022a) might be a key for practical interactive agents. Since our approach – taking raw HTML and screenshots as inputs and predicting parsable actions in text – only has minimal assumptions which constraint model architectures, it might be applicable to any advanced LLMs or open-ended situations. While WebGUM could deal with out-of-distribution compositional and perturbed tasks in a robust manner, human-level broader generalization to the diverse real websites or instructions is still a hard problem to be resolved.

## 7 CONCLUSION

We develop *Web navigation via Grounded Understanding Models* (WebGUM), learning an instruction-following visual language foundation model for web navigation. WebGUM significantly improves the success rate on MiniWoB, compared to previous offline-trained SoTA from 48.4% to 94.2%. Our detailed ablations show that temporal and local visual tokens capture dynamic transition and visual context of the page, and that instruction-finetuned language models significantly improves web navigation performance due to the better HTML comprehension and capability of multi-step reasoning. Multi-step reasoning enables more robust generalization to out-of-distribution tasks, and outperforms PaLM-540B in WebShop. WebGUM also demonstrates strong positive transfer to real-world action prediction tasks in Mind2Web. Furthermore, we scale the existing MiniWoB dataset into multimodal 347K expert demonstrations, about 38 times larger than before. We believe that our work is an significant step towards building more capable and scalable models for autonomous web navigation.

ACKNOWLEDGEMENTS

HF was supported by JSPS KAKENHI Grant Number JP22J21582. We thank Yusuke Iwasawa, Mustafa Safdari, Austin Huang, Heiga Zen for helpful feedback on this work, and Shunyu Yao for setting up WebShop experiments.

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

APPENDIX

## A    BROADER IMPACTS

While WebGUM is evaluated only in realistic web simulators (Shi et al., 2017; Liu et al., 2018; Yao et al., 2022a), we should carefully conduct it if we deploy the autonomous web agent on the real-world Internet because of security and safety reasons. For instance, the wrong password may cause an account freeze, and emailing the wrong person is problematic in a business scene. Training with online RL may often be infeasible for this reason, while we demonstrate an alternative approach; data-driven, fully offline training by leveraging inductive bias in foundation models. Autonomous agents, well-grounded with the user's intention, should be helpful in our daily lives by reducing our burden on computer tasks. Because a part of our training corpus (54K) includes the demonstrations taken from the output of LLMs (Anil et al., 2023), we will exclude those from the dataset release and it will result in 347K episodes.

## B    EXTENDED RELATED WORKS

**Foundation Models for Decision Making**  Recently, the ability of multi-step reasoning and inductive bias in foundation models have been leveraged to solve text-based interactive tasks via sequential decisions considering few-shot in-context examples (Ahn et al., 2022; Huang et al., 2022a;b; Zeng et al., 2022; Yao et al., 2022b; Meta Fundamental AI Research Diplomacy Team et al., 2022). Even in continuous control (Chen et al., 2021a; Janner et al., 2021; Furuta et al., 2022b; Brohan et al., 2022) or computer games (Reed et al., 2022; Lee et al., 2022b; Fan et al., 2022), high-capacity transformer models are trained with a large amount of diverse dataset via multi-task behavioral distillation (Chen et al., 2021c; Gu et al., 2021a; DeepMind Interactive Agents Team et al., 2021; Furuta et al., 2022a; Shridhar et al., 2022; Jiang et al., 2022). To build autonomous web navigation agents, we also leverage pre-trained LLM (Raffel et al., 2020; Chung et al., 2022), by finetuning with massively-curated multimodal demonstrations, and we point out that the better content comprehension and multi-step reasoning abilities, obtained through instruction-finetuning of LLM (Chung et al., 2022), are essential for the notable performance on downstream decision making aligned with human instructions.

**Multimodal Large-scale Models**  Large language models have demonstrated extraordinary emergent abilities on a variety of NLP tasks, such as commonsense question answering, arithmetic, logical reasoning, open-ended text generation (Radford et al., 2019; Brown et al., 2020; Chowdhery et al., 2022; Wei et al., 2022b; Tay et al., 2022), or code completion (Chen et al., 2021b; Austin et al., 2021; Li et al., 2022b). In addition, some works have investigated vision-and-language understanding to improve the accuracy of common vision-based tasks such as open-ended image/object classification (Radford et al., 2021; Gu et al., 2021b; Kamath et al., 2021), image captioning, or visual question-answering (Lu et al., 2022; Alayrac et al., 2022; Chen et al., 2022; Reed et al., 2022; Liu et al., 2023; Dai et al., 2023; Li et al., 2023). Several works also have tackled document understanding with (multimodal) transformer models (Xu et al., 2019; Li et al., 2021a;c; Appalaraju et al., 2021; Tang et al., 2022; Wang et al., 2022a;b), including markup languages such as HTML (Aghajanyan et al., 2021; 2022; Li et al., 2021b; Lee et al., 2022a) for summarization of the documents or question answering on the contents. Despite the great efforts on document understanding, these works are less connected to interactive decision making problems. Our model obtains not only a grounded understanding of websites in a multimodal manner but also the ability to decide the optimal actions to achieve given instructions in web navigation, helping multi-step decisions and visual context understanding.

## C    IMPLEMENTATION DETAILS

We adopt the encoder-decoder models proposed by Raffel et al. (2020) as multimodal transformers, and vision transformer (Dosovitskiy et al., 2020) pre-trained with ImageNet-21K (Deng et al., 2009) as an image encoder for the visual tokens[3]. We especially use ViT-B16, a small-size transformer with 86 million parameters, which divides an input image into $16 \times 16$-size patches. We use publicly

---

[3]https://github.com/google-research/scenic

available checkpoints of T5 (Raffel et al., 2020)[4], Flan-T5 (Chung et al., 2022)[5], and T5-XL finetuned with MiniWoB++ demonstrations (Gur et al., 2022)[6] for the experiments. To construct the training pipeline, we leverage SeqIO (Roberts et al., 2022) library, and use SentencePiece (Kudo & Richardson, 2018) vocabulary with 32K tokens from C4 dataset (Raffel et al., 2020) for text tokenization. The batch size for training is 128, and input sequence length is set to 4096 tokens. Due to the huge computational requirements, we run one seed to train each model throughout the paper (Humphreys et al., 2022; Gur et al., 2022). We use cloud TPU-v4, which has a 32 GiB HBM memory space for the experiments. Base-size models require 256 cores and XL-size models do 512 cores, which takes 1-2 days for finetuning.

# D    DETAILS ON DATASET AND MODEL SIZE SCALING

We here present how critical it is to scale up the dataset and model size in WebGUM. For the dataset size ablation, we use Flan-T5-Base and ViT-B16. As for both HTML and multimodal models, we could observe the scaling effects in web navigation: the larger the dataset (Table 4) and model (Table 5) size are, the higher the success rates are. Surprisingly, our approach even with only 2.8K HTML episodes (about 25% of the previous one curated by Liu et al. (2018)) and Base-size model (about 7.3% parameters) already achieves 55.7%, surpassing previous SL state-of-the-art (48.4% by Gur et al. (2022)). This surprising efficiency might come from the sufficient inductive bias and alignment with the user intentions in instruction-finetuned LLMs, and WebGUM could fully leverage them for web automation problems. The margin of improvement might be smaller than expected due to the limited capacity of transformer to obtain the grounded understanding of natural language instructions, HTML, and screenshots. In fact, the results also reveal that scaling the models might be more important than the dataset; the low-capacity model may cap the performance at a lower level.

| Pre-Trained Models | Modality | Dataset | Success Rate |
|---|---|---|---|
| T5-XL (Gur et al., 2022) | HTML | 12K | 48.4% |
| T5-XL | HTML | 347K | 49.8% |
| Flan-T5-Base | HTML | 2.8K | 55.7% |
| Flan-T5-Base | HTML | 68K | 56.3% |
| Flan-T5-Base | HTML | 347K | 57.2% |
| Flan-T5-Base, ViT-B16 | HTML+Image | 2.8K | 61.1% |
| Flan-T5-Base, ViT-B16 | HTML+Image | 68K | 62.3% |
| Flan-T5-Base, ViT-B16 | HTML+Image | 347K | 66.1% |

Table 4: Average success rate of WebGUM with different dataset sizes. We observe the larger the dataset size is, the higher the success rate is. Surprisingly, our approach outperforms previous state-of-the-art by over 7.3% even with 2.8K-episode dataset (about 25% of the previous dataset curated by Liu et al. (2018)).

| Pre-Trained Models | # of Params | Modality | Success Rate |
|---|---|---|---|
| Flan-T5-Base | 220M | HTML | 57.2% |
| Flan-T5-Large | 770M | HTML | 72.4% |
| Flan-T5-XL | 3B | HTML | 75.5% |
| Flan-T5-XXL | 11B | HTML | 79.0% |
| Flan-T5-Base, ViT-B16 | 310M | HTML+Image | 66.1% |
| Flan-T5-Large, ViT-B16 | 860M | HTML+Image | 77.4% |
| Flan-T5-XL, ViT-B16 | 3B | HTML+Image | 80.3% |

Table 5: Average success rate of WebGUM with different model sizes. As for both HTML-only and multimodal models, we could observe the performance increases as the model size does.

---

[4] https://github.com/google-research/t5x/blob/main/docs/models.md#t5-11-checkpoints

[5] https://github.com/google-research/t5x/blob/main/docs/models.md#flan-t5-checkpoints

[6] https://console.cloud.google.com/storage/browser/gresearch/webllm/webn_t5_3b

# E  WEBSRC

We extensively evaluate the capability of HTML comprehension in instruction-finetuned LLMs with WebSRC (Chen et al., 2021d) where the models are asked to solve contextual QA problems understanding a given HTML and its structure. Those problems are curated from real websites to include key-value extraction, entity comparison, and table understanding problems. The answer formats are either text span in HTML or binary (yes/no). Because the context length is insufficient for raw HTML, we preprocess context HTML by extracting a snippet that includes the answers in advance. We finetune both T5-XL and Flan-T5-XL with the training dataset. Table 6 shows that Flan-T5 records better HTML comprehension performance than T5, which may accelerates the web navigation performance on MiniWoB++ and Mind2Web.

| Models | EM | F1 |
|---|---|---|
| T5-XL | 63.85 | 71.44 |
| Flan-T5-XL | **68.91** | **78.48** |

Table 6: Base language model performance in WebSRC (Chen et al., 2021d). We finetune both T5 and Flan-T5 with trainng dataset. Flan-T5 achieves better performance in HTML comprehension than T5.

# F  DATASET DETAILS

To construct a large-scale multimodal behavioral dataset on MiniWoB++, we leverage a public finetuned-LLM policy (Gur et al., 2022) trained with multi-task human demonstration dataset (Liu et al., 2018)[7] as a demonstrator. We run such LLM policies with 10,000 episodes per task and only keep successful trajectories to maintain the quality of dataset, following Humphreys et al. (2022). Lastly, we collect additional 54K demonstrations with Synapse (Zheng et al., 2023)[8], a private-LLM-based agents with prompting, for the tasks where finetuned-LLM may not complete well such as `click-scroll-list` and `enter-time`, and also write a scripted policy for `book-flight`. We use PaLM 2 (Anil et al., 2023) as a base LLM for Synapse. Such efforts result in a multi-task dataset with 401K (347K+54K) episodes including HTML and screenshots at each time step. Table 7 shows the details of our multimodal dataset (347K), consisting of HTML, screenshots, actions, and instructions at each time step.

---

[7]https://github.com/stanfordnlp/miniwob-plusplus-demos
[8]https://github.com/ltzheng/synapse

| Task | # of episodes | # of steps | Ratio (episode) |
|---|---|---|---|
| book-flight | 9999 | 90177 | 2.88% |
| choose-date | 383 | 1508 | 0.11% |
| choose-date-easy | 3353 | 12946 | 0.97% |
| choose-date-medium | 2222 | 8733 | 0.64% |
| choose-list | 1861 | 3724 | 0.54% |
| click-button | 9782 | 9909 | 2.82% |
| click-button-sequence | 10000 | 20000 | 2.88% |
| click-checkboxes | 9761 | 28904 | 2.81% |
| click-checkboxes-large | 1962 | 19072 | 0.57% |
| click-checkboxes-soft | 9228 | 36384 | 2.66% |
| click-checkboxes-transfer | 10000 | 59793 | 2.88% |
| click-collapsible | 5947 | 13077 | 1.71% |
| click-collapsible-2 | 2199 | 5627 | 0.63% |
| click-color | 2554 | 2554 | 0.74% |
| click-dialog | 10000 | 10000 | 2.88% |
| click-dialog-2 | 3285 | 3285 | 0.95% |
| click-link | 9961 | 9961 | 2.87% |
| click-menu | 3238 | 3243 | 0.93% |
| click-option | 9998 | 20000 | 2.88% |
| click-pie | 3724 | 8548 | 1.07% |
| click-scroll-list | 0 | 0 | 0.00% |
| click-shades | 0 | 0 | 0.00% |
| click-shape | 6116 | 6117 | 1.76% |
| click-tab | 9978 | 13177 | 2.88% |
| click-tab-2 | 1844 | 2109 | 0.53% |
| click-tab-2-hard | 1574 | 1916 | 0.45% |
| click-test | 10000 | 10000 | 2.88% |
| click-test-2 | 10000 | 10000 | 2.88% |
| click-widget | 9963 | 9963 | 2.87% |
| count-shape | 5849 | 5893 | 1.69% |
| email-inbox | 5159 | 14258 | 1.49% |
| email-inbox-forward-nl | 9995 | 39980 | 2.88% |
| email-inbox-forward-nl-turk | 4900 | 20165 | 1.41% |
| email-inbox-nl-turk | 4346 | 11416 | 1.25% |
| enter-date | 10000 | 20000 | 2.88% |
| enter-password | 9980 | 29940 | 2.88% |
| enter-text | 10000 | 20000 | 2.88% |
| enter-text-dynamic | 9983 | 19966 | 2.88% |
| enter-time | 0 | 0 | 0.00% |
| focus-text | 10000 | 10000 | 2.88% |
| focus-text-2 | 10000 | 10000 | 2.88% |
| grid-coordinate | 8353 | 8353 | 2.41% |
| guess-number | 1021 | 2042 | 0.29% |
| identify-shape | 9007 | 9010 | 2.60% |
| login-user | 9793 | 29379 | 2.82% |
| login-user-popup | 9786 | 39170 | 2.82% |
| multi-layouts | 10000 | 40000 | 2.88% |
| multi-orderings | 10000 | 40000 | 2.88% |
| navigate-tree | 9864 | 15140 | 2.84% |
| search-engine | 8872 | 35095 | 2.56% |
| social-media | 2631 | 4407 | 0.76& |
| social-media-all | 95 | 208 | 0.03% |
| social-media-some | 319 | 893 | 0.09& |
| tic-tac-toe | 3947 | 13773 | 1.14% |
| use-autocomplete | 3465 | 6930 | 1.00% |
| use-spinner | 530 | 532 | 0.15% |
| **Total** | 346827 | 867277 | 100% |

Table 7: Details of our multimodal dataset. It contains about 347K episodes in total.

# G   PER-TASK PERFORMANCE OF MINIWOB++

In this section, we present per-task success rate on MiniWoB++ (Table 8) and absolute performance improvement by adding image modality to HTML input for WebGUM (Figure 7).

As for Table 8, we refer to Gur et al. (2022) and Zheng et al. (2023) for the baseline performances. We use 56 tasks as benchmark, while removing some duplicated tasks (e.g. "-nodelay" tasks) from 62 tasks adopted in Gur et al. (2022). During the evaluation on MiniWoB++, we ignore the time limit due to the computational constraints.

Figure 7 presents full results of the absolute performance improvement, subtracting the success rates: (Success Rate of WebGUM(HTML+Image)) − (Success Rate of WebGUM(HTML)). The results suggest WebGUM leverages visual inputs for multi-step tasks with dynamic page transitions (e.g. book-flight or search-engine) or the tasks that require global contexts of the page (e.g. tic-tac-toe or click-shape). See Appendix L for the visualization.

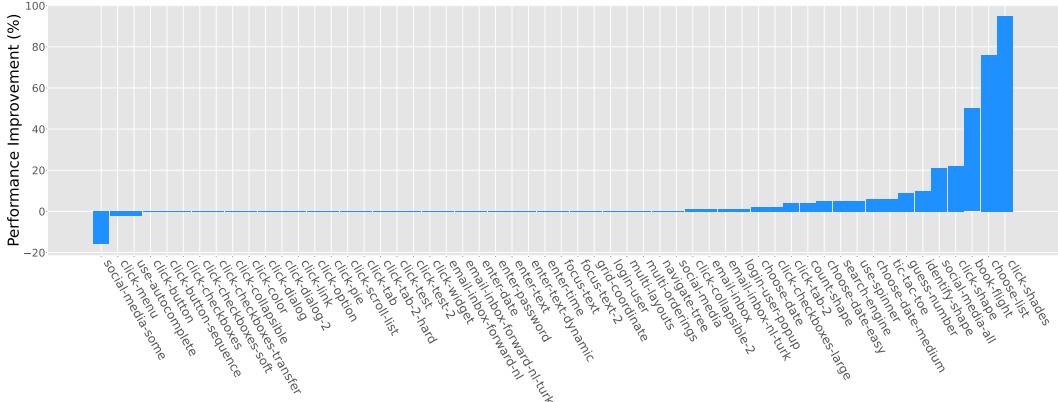

Figure 7: Performance improvement by adding image modality to HTML on 56 tasks from MiniWoB++. We subtract the success rates: (Success Rate of WebGUM(HTML+Image)) − (Success Rate of WebGUM(HTML)).

| Task | Synapse | Human | AdaPlanner | RCI | RCI (GPT-4) | CC-Net | CC-Net (SL) | WGE | WebN-T5 | WebGUM (HTML) | WebGUM |
|---|---|---|---|---|---|---|---|---|---|---|---|
| bisect-angle | n/a | 0.92 | n/a | n/a | n/a | 0.97 | 0.29 | n/a | n/a | n/a | n/a |
| book-flight | 0.76 | 0.87 | n/a | n/a | n/a | 0.87 | 0.00 | 0.00 | 0.00 | 0.48 | 0.98 |
| chase-circle | n/a | 0.82 | n/a | n/a | n/a | 0.93 | 0.80 | n/a | n/a | n/a | n/a |
| choose-date | 1.00 | 0.97 | n/a | n/a | n/a | 0.97 | 0.12 | 0.00 | 0.00 | 0.98 | 1.00 |
| choose-date-easy | n/a | 0.99 | n/a | n/a | n/a | 0.99 | 0.42 | n/a | 0.03 | 0.95 | 1.00 |
| choose-date-medium | n/a | 0.98 | n/a | n/a | n/a | 0.99 | 0.26 | n/a | 0.00 | 0.94 | 1.00 |
| choose-list | 1.00 | 0.98 | 1.00 | 1.00 | 1.00 | 0.99 | 0.19 | 0.16 | 0.26 | 0.24 | 1.00 |
| circle-center | n/a | 0.96 | n/a | n/a | n/a | 0.97 | 0.36 | n/a | n/a | n/a | n/a |
| click-button | 1.00 | 0.98 | 1.00 | 1.00 | 1.00 | 1.00 | 0.78 | 1.00 | 1.00 | 1.00 | 1.00 |
| click-button-sequence | 1.00 | 0.94 | 1.00 | 1.00 | 1.00 | 1.00 | 0.47 | 0.99 | 1.00 | 1.00 | 1.00 |
| click-checkboxes | 1.00 | 0.97 | 1.00 | 1.00 | 1.00 | 0.98 | 0.32 | 0.98 | 0.96 | 1.00 | 1.00 |
| click-checkboxes-large | 1.00 | 0.87 | 1.00 | 0.94 | 0.94 | 0.71 | 0.00 | 0.68 | 0.22 | 0.97 | 0.99 |
| click-checkboxes-soft | 1.00 | 0.73 | 0.80 | 0.72 | 0.96 | 0.95 | 0.04 | 0.51 | 0.54 | 1.00 | 1.00 |
| click-checkboxes-transfer | 1.00 | 0.98 | 0.98 | 1.00 | 1.00 | 0.99 | 0.36 | 0.64 | 0.63 | 1.00 | 1.00 |
| click-collapsible | 1.00 | 0.99 | 1.00 | 1.00 | 1.00 | 1.00 | 0.81 | 1.00 | 0.00 | 1.00 | 1.00 |
| click-collapsible-2 | 0.96 | 0.97 | 0.84 | 0.62 | 1.00 | 0.98 | 0.17 | 0.65 | 0.00 | 0.94 | 0.95 |
| click-color | 1.00 | 0.97 | 1.00 | 1.00 | 1.00 | 1.00 | 0.82 | 1.00 | 0.27 | 1.00 | 1.00 |
| click-dialog | 1.00 | 1.00 | 1.00 | 1.00 | 1.00 | 1.00 | 0.95 | 1.00 | 1.00 | 1.00 | 1.00 |
| click-dialog-2 | 1.00 | 0.99 | 1.00 | 1.00 | 1.00 | 1.00 | 0.88 | 1.00 | 0.24 | 1.00 | 1.00 |
| click-link | 1.00 | 0.99 | 0.98 | 1.00 | 1.00 | 0.99 | 0.59 | 1.00 | 1.00 | 1.00 | 1.00 |
| click-menu | 1.00 | 0.97 | 0.78 | 1.00 | 1.00 | 0.94 | 0.22 | n/a | 0.37 | 0.99 | 0.97 |
| click-menu-2 | n/a | 0.98 | n/a | n/a | n/a | 0.83 | 0.52 | n/a | n/a | n/a | n/a |
| click-option | 1.00 | 0.99 | 1.00 | 1.00 | 1.00 | 0.99 | 0.21 | 1.00 | 0.87 | 1.00 | 1.00 |
| click-pie | 1.00 | 0.98 | n/a | n/a | n/a | 0.97 | 0.15 | 0.32 | 0.51 | 0.99 | 0.99 |
| click-scroll-list | 1.00 | 0.91 | 1.00 | 1.00 | 1.00 | 0.60 | 0.01 | n/a | 0.00 | 1.00 | 1.00 |
| click-shades | 1.00 | 0.91 | 1.00 | 1.00 | 1.00 | 1.00 | 0.04 | 0.22 | 0.00 | 0.05 | 1.00 |
| click-shape | 0.96 | 0.88 | 0.75 | 0.98 | 0.98 | 0.95 | 0.11 | 0.64 | 0.53 | 0.72 | 0.94 |
| click-tab | 1.00 | 0.99 | 1.00 | 1.00 | 1.00 | 1.00 | 0.95 | 0.55 | 0.74 | 1.00 | 1.00 |
| click-tab-2 | 0.94 | 0.97 | 0.85 | 0.74 | 1.00 | 0.98 | 0.27 | 0.64 | 0.18 | 0.95 | 0.99 |
| click-tab-2-easy | n/a | 0.99 | n/a | n/a | n/a | 0.99 | 0.61 | n/a | n/a | n/a | n/a |
| click-tab-2-hard | 0.96 | 0.96 | 0.78 | 0.76 | 0.98 | 0.98 | 0.19 | n/a | 0.12 | 0.95 | 0.95 |
| click-tab-2-medium | n/a | 0.97 | n/a | n/a | n/a | 0.99 | 0.54 | n/a | n/a | n/a | n/a |
| click-test | 1.00 | 1.00 | 1.00 | 1.00 | 1.00 | 1.00 | 1.00 | 1.00 | 1.00 | 1.00 | 1.00 |
| click-test-2 | 1.00 | 0.99 | 1.00 | 1.00 | 1.00 | 1.00 | 0.95 | 1.00 | 1.00 | 1.00 | 1.00 |

| | | | | | | | | | | | |
|---|---|---|---|---|---|---|---|---|---|---|---|
| click-test-transfer | n/a | 0.99 | n/a | n/a | n/a | 1.00 | 0.94 | n/a | n/a | n/a | n/a |
| click-widget | 1.00 | 0.83 | 1.00 | 0.98 | 0.98 | 1.00 | 0.56 | 0.93 | 1.00 | 1.00 | 1.00 |
| copy-paste | 1.00 | 0.94 | n/a | n/a | n/a | 0.79 | 0.04 | n/a | n/a | n/a | n/a |
| copy-paste-2 | 1.00 | 0.94 | n/a | n/a | n/a | 0.63 | 0.01 | n/a | n/a | n/a | n/a |
| count-shape | 0.78 | 0.82 | 0.50 | 0.40 | 0.4 | 0.85 | 0.21 | 0.59 | 0.41 | 0.64 | 0.68 |
| count-sides | n/a | 0.98 | n/a | n/a | n/a | 1.00 | 0.74 | n/a | n/a | n/a | n/a |
| drag-box | n/a | 0.99 | n/a | n/a | n/a | 1.00 | 0.61 | n/a | n/a | n/a | n/a |
| drag-cube | n/a | 0.99 | n/a | n/a | n/a | 0.79 | 0.23 | n/a | n/a | n/a | n/a |
| drag-item | n/a | 0.98 | n/a | n/a | n/a | 1.00 | 0.61 | n/a | n/a | n/a | n/a |
| drag-items | n/a | 0.93 | n/a | n/a | n/a | 0.99 | 0.13 | n/a | n/a | n/a | n/a |
| drag-items-grid | n/a | 0.87 | n/a | n/a | n/a | 0.98 | 0.05 | n/a | n/a | n/a | n/a |
| drag-shapes | n/a | 0.96 | n/a | n/a | n/a | 0.99 | 0.26 | n/a | n/a | n/a | n/a |
| drag-sort-numbers | n/a | 0.92 | n/a | n/a | n/a | 0.97 | 0.11 | n/a | n/a | n/a | n/a |
| email-inbox | 1.00 | 0.96 | 0.98 | 0.98 | 0.98 | 1.00 | 0.09 | 0.43 | 0.38 | 0.99 | 1.00 |
| email-inbox-delete | n/a | 0.99 | n/a | n/a | n/a | 1.00 | 0.22 | n/a | n/a | n/a | n/a |
| email-inbox-forward | n/a | 0.96 | n/a | n/a | n/a | 1.00 | 0.01 | n/a | n/a | n/a | n/a |
| email-inbox-forward-nl | 1.00 | 0.91 | 1.00 | 1.00 | 1.00 | 1.00 | 0.00 | n/a | 0.60 | 1.00 | 1.00 |
| email-inbox-forward-nl-turk | 1.00 | 0.88 | 1.00 | 0.94 | 0.94 | 1.00 | 0.00 | n/a | 0.33 | 1.00 | 1.00 |
| email-inbox-important | n/a | 0.99 | n/a | n/a | n/a | 1.00 | 0.30 | n/a | n/a | n/a | n/a |
| email-inbox-nl-turk | 1.00 | 0.93 | 0.90 | 0.98 | 0.98 | 1.00 | 0.05 | 0.77 | 0.23 | 0.99 | 1.00 |
| email-inbox-noscroll | n/a | 0.96 | n/a | n/a | n/a | 1.00 | 0.13 | | n/a | n/a | n/a |
| email-inbox-reply | n/a | 0.91 | n/a | n/a | n/a | 1.00 | 0.00 | n/a | n/a | n/a | n/a |
| email-inbox-star-reply | n/a | 0.95 | n/a | n/a | n/a | 1.00 | 0.11 | n/a | n/a | n/a | n/a |
| enter-date | 1.00 | 0.97 | 1.00 | 0.96 | 0.96 | 1.00 | 0.02 | 0.00 | 0.00 | 1.00 | 1.00 |
| enter-password | 1.00 | 0.96 | 0.98 | 1.00 | 1.00 | 1.00 | 0.02 | 0.99 | 0.97 | 1.00 | 1.00 |
| enter-text | 1.00 | 0.98 | 0.98 | 1.00 | 1.00 | 1.00 | 0.35 | 1.00 | 0.89 | 1.00 | 1.00 |
| enter-text-2 | n/a | 0.91 | n/a | n/a | n/a | 0.98 | 0.04 | n/a | n/a | n/a | n/a |
| enter-text-dynamic | 1.00 | 0.97 | 0.96 | 1.00 | 1.00 | 1.00 | 0.39 | 1.00 | 0.98 | 1.00 | 1.00 |
| enter-time | 0.98 | 0.98 | 0.96 | 1.00 | 1.00 | 0.97 | 0.04 | 0.52 | 0.00 | 1.00 | 1.00 |
| find-midpoint | n/a | 0.94 | n/a | n/a | n/a | 0.97 | 0.35 | n/a | n/a | n/a | n/a |
| find-word | 0.84 | 0.96 | n/a | n/a | n/a | 0.88 | 0.05 | n/a | n/a | n/a | n/a |
| focus-text | 1.00 | 1.00 | 1.00 | 1.00 | 1.00 | 1.00 | 0.99 | 1.00 | 1.00 | 1.00 | 1.00 |
| focus-text-2 | 1.00 | 0.99 | 0.94 | 1.00 | 1.00 | 1.00 | 0.96 | 1.00 | 1.00 | 1.00 | 1.00 |
| grid-coordinate | 1.00 | 0.87 | 1.00 | 1.00 | 1.00 | 1.00 | 0.66 | 1.00 | 0.49 | 1.00 | 1.00 |
| guess-number | 1.00 | 0.99 | 0.88 | 0.20 | 0.20 | 1.00 | 0.21 | 0.00 | 0.00 | 0.34 | 0.43 |
| highlight-text | n/a | 0.97 | n/a | n/a | n/a | 1.00 | 0.51 | n/a | n/a | n/a | n/a |
| highlight-text-2 | n/a | 0.97 | n/a | n/a | n/a | 1.00 | 0.40 | n/a | n/a | n/a | n/a |
| identify-shape | 1.00 | 0.98 | 0.96 | 0.76 | 1.0 | 1.00 | 0.68 | 0.90 | 0.88 | 0.90 | 1.00 |
| login-user | 1.00 | 0.96 | 1.00 | 1.00 | 1.0 | 1.00 | 0.00 | 0.99 | 0.82 | 1.00 | 1.00 |
| login-user-popup | 1.00 | 0.94 | 0.98 | 0.68 | 0.68 | 1.00 | 0.02 | n/a | 0.72 | 0.99 | 1.00 |
| moving-items | n/a | 0.18 | n/a | n/a | n/a | 0.88 | 0.13 | n/a | n/a | n/a | n/a |
| multi-layouts | 0.94 | 0.95 | 0.84 | 0.72 | 0.96 | 1.00 | 0.00 | 0.99 | 0.83 | 1.00 | 1.00 |
| multi-orderings | 1.00 | 0.96 | 1.00 | 1.00 | 1.00 | 1.00 | 0.00 | 0.99 | 0.88 | 1.00 | 1.00 |
| navigate-tree | 0.96 | 0.98 | 0.82 | 0.86 | 1.00 | 0.99 | 0.32 | 0.99 | 0.91 | 1.00 | 1.00 |
| number-checkboxes | n/a | 0.96 | n/a | n/a | n/a | 0.99 | 0.00 | n/a | n/a | n/a | n/a |
| read-table | 1.00 | 0.97 | n/a | n/a | n/a | 0.97 | 0.01 | n/a | n/a | n/a | n/a |
| read-table-2 | n/a | 0.95 | n/a | n/a | n/a | 0.94 | 0.00 | n/a | n/a | n/a | n/a |
| resize-textarea | n/a | 0.94 | n/a | n/a | n/a | 1.00 | 0.27 | n/a | n/a | n/a | n/a |
| right-angle | n/a | 0.87 | n/a | n/a | n/a | 0.98 | 0.26 | n/a | n/a | n/a | n/a |
| scroll-text | n/a | 0.97 | n/a | n/a | n/a | 0.96 | 0.04 | n/a | n/a | n/a | n/a |
| scroll-text-2 | n/a | 0.97 | n/a | n/a | n/a | 1.00 | 0.88 | n/a | n/a | n/a | n/a |
| search-engine | 1.00 | 0.97 | 1.00 | 1.00 | 1.00 | 1.00 | 0.15 | 0.26 | 0.34 | 0.91 | 0.96 |
| simon-says | n/a | 0.62 | n/a | n/a | n/a | 0.00 | 0.02 | n/a | n/a | n/a | n/a |
| simple-algebra | 1.00 | 0.86 | 0.82 | 1.00 | 1.00 | 0.75 | 0.03 | n/a | n/a | n/a | n/a |
| simple-arithmetic | 1.00 | 0.96 | n/a | n/a | 1.00 | 0.86 | 0.38 | n/a | n/a | n/a | n/a |
| social-media | 1.00 | 0.96 | 0.82 | 0.98 | 0.98 | 0.90 | 0.03 | 0.39 | 0.21 | 1.00 | 1.00 |
| social-media-all | 1.00 | 0.89 | 1.00 | 1.00 | 1.00 | 0.75 | 0.00 | 0.01 | 0.00 | 0.31 | 0.52 |
| social-media-some | 1.00 | 0.91 | 0.90 | 0.90 | 0.96 | 0.85 | 0.01 | 0.01 | 0.02 | 0.89 | 0.73 |
| terminal | 0.98 | 0.88 | 0.98 | 1.00 | 1.00 | 0.00 | 0.00 | n/a | n/a | n/a | n/a |
| text-editor | n/a | 0.88 | n/a | n/a | n/a | 0.98 | 0.11 | n/a | n/a | n/a | n/a |
| text-transform | 1.00 | 0.86 | n/a | 0.80 | 0.80 | 0.60 | 0.19 | n/a | n/a | n/a | n/a |
| tic-tac-toe | 1.00 | 0.71 | 0.48 | 0.56 | 0.56 | 0.83 | 0.32 | 0.37 | 0.48 | 0.50 | 0.56 |
| unicode-test | 1.00 | 0.99 | n/a | n/a | n/a | 1.00 | 0.86 | n/a | n/a | n/a | n/a |
| use-autocomplete | 0.98 | 0.98 | 0.88 | 0.58 | 0.58 | 1.00 | 0.07 | 0.78 | 0.22 | 1.00 | 0.98 |
| use-colorwheel | n/a | 0.90 | n/a | n/a | n/a | 0.98 | 0.68 | n/a | n/a | n/a | n/a |
| use-colorwheel-2 | n/a | 0.94 | n/a | n/a | n/a | 0.95 | 0.38 | n/a | n/a | n/a | n/a |
| use-slider | n/a | 0.98 | n/a | n/a | n/a | 0.91 | 0.18 | n/a | n/a | n/a | n/a |
| use-slider-2 | n/a | 0.97 | n/a | n/a | n/a | 0.95 | 0.03 | n/a | n/a | n/a | n/a |
| use-spinner | 1.00 | 0.98 | 0.90 | 0.88 | 0.96 | 1.00 | 0.47 | 0.04 | 0.07 | 0.06 | 0.11 |
| visual-addition | n/a | 0.97 | n/a | n/a | n/a | 0.99 | 0.36 | n/a | n/a | n/a | n/a |
| **Average** | 0.985 | 0.935 | 0.929 | 0.906 | 0.940 | 0.935 | 0.305 | 0.646 | 0.484 | 0.887 | 0.942 |
| **# of Tasks** | 63 | 104 | 53 | 54 | 54 | 104 | 104 | 48 | 56 | 56 | 56 |

Table 8: Per-task success rate on MiniWoB++. We refer to Gur et al. (2022) and Zheng et al. (2023) for the baseline performances.

## H   COMPOSITIONAL EVALUATION ON MINIWOB++

For the compositional evaluation, we pick up 4 `click-`"something" (link, button, checkboxes, dialog) tasks and make some combinations of those by naively stitching with 2 or 3 tasks. Then, we prepare the following 6 combinational tasks,

- `click-button_click-checkboxes`
- `click-button_click-dialog`
- `click-button_click-link`
- `click-link_click-button`
- `click-link_click-button_click-dialog`
- `click-link_click-dialog`

These tasks should be resolved in order of the name: for instance, in `click-link_click-button_click-dialog` task, the agent should click the proper link, click the proper button, click the proper dialog, and then the task results in success. In `click-button_click-link` task, the agent should click the proper button, and then click the proper link. The instructions for compositional tasks are also simply combined among original task instructions in order of the name. This evaluation could test the ability to transfer primitive skills to control computers to solve unseen tasks. Table 9 shows the per-task average success rate among 6 combinations above. WebGUM can solve the compositional tasks much better than baselines (Gur et al., 2022; Zheng et al., 2023) .

| Compositional Task | WebN-T5 | Synapse | WebGUM (HTML) | WebGUM (HTML+Image) |
|---|---|---|---|---|
| click-button_click-checkboxes | 0.26 | 0.84 | 0.21 | 0.27 |
| click-button_click-dialog | 0.95 | 1.00 | 0.87 | 0.93 |
| click-button_click-link | 0.87 | 0.99 | 0.81 | 0.88 |
| click-link_click-button | 0.35 | 1.00 | 0.90 | 0.95 |
| click-link_click-button_click-dialog | 0.08 | 0.60 | 0.73 | 0.73 |
| click-link_click-dialog | 0.55 | 0.00 | 0.93 | 0.95 |
| **Ave.** | 0.510 | 0.738 | 0.742 | 0.785 |

Table 9: Per-task average success rate on 6 tasks from compositional MiniWoB++.

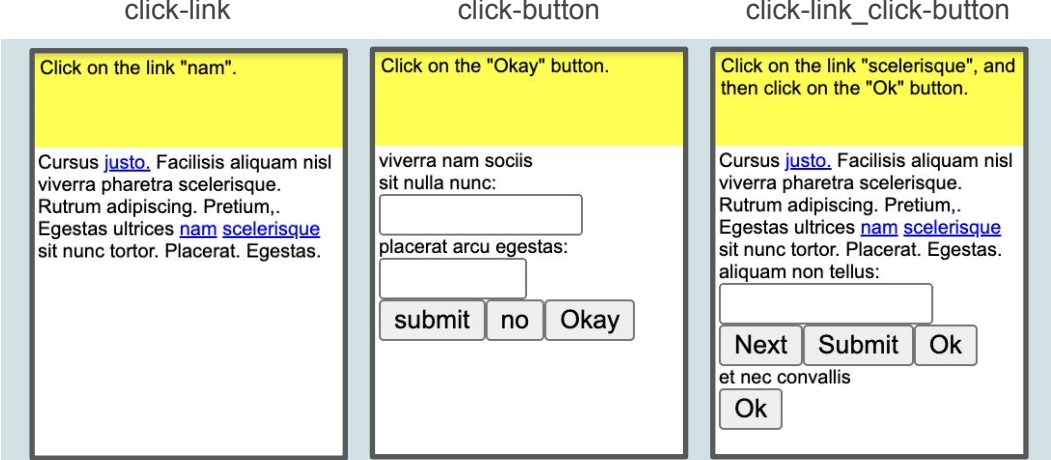

Figure 8: Example of compositional evaluation on MiniWoB++ (the same as Figure 5).

# I    COMPARISON AGAINST PRIOR WEB NAVIGATION AGENTS

| Methods | Architecture | Pre-trained | Input | Output | Offline |
|---|---|---|---|---|---|
| WGE (Liu et al., 2018) | LSTM, self-attention | ✗ | DOM | Logit of action | ✗ |
| CoDE (Gur et al., 2019; 2021) | Bi-LSTM | ✗ | DOM | Logit of action | ✗ |
| DOM-Q-NET(Jia et al., 2019) | GNN | ✗ | DOM | Logit of action | ✗ |
| CC-Net (Humphreys et al., 2022) | LSTM, Transformer, ResNet | ✗* | DOM, Screenshot | Logit of action | ✗ |
| WebShop (Yao et al., 2022a) | BERT, BART | ✔ | Text (from HTML) | Logit of action | ✗ / ✔ |
| WebGUM (Ours) | T5 Transformer, ViT | ✔ | HTML, Screenshot | Text | ✔ |

Table 10: Prior works have studied web navigation problem as online RL to learn the optimal action distribution with task-specific model architectures from scratch (*or partially using pre-trained vision encoder). We omit the web-specialized architecture and input-output space, and convert web navigation into visual question-answering format (text, image → text), which allows us to learn the agents offline by leveraging pre-trained foundation models (Raffel et al., 2020; Chung et al., 2022; Dosovitskiy et al., 2020) in vision or language domains as strong inductive bias for web environments.

# J    INPUT PERTURBATION EVALUATION ON MINIWOB++

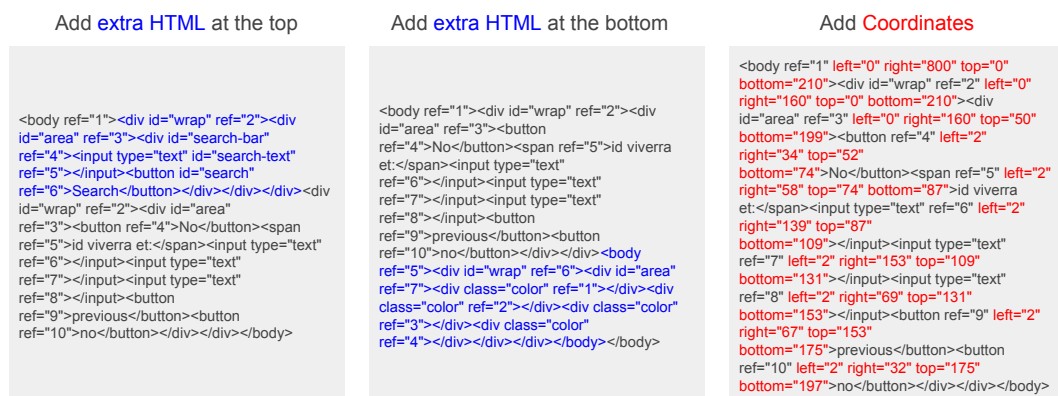

Figure 9: Example of input perturbation for MiniWoB++ evaluation (the same as Figure 6).

# K    EVALUATION ON WEBSHOP

In addition to MiniWoB++, we extensively evaluate our WebGUM on WebShop (Yao et al., 2022a) benchmark, another online-shopping websites simulator with a large amount of real-world product data. WebShop provides user instruction that describes the feature of items (e.g. *I need a long clip-in hair extension which is natural looking, and price lower than 20.00 dollars*). The agents should search, compare and choose a proper product that matches the given instruction. Since WebShop requires complex multi-step reasoning considering previous contexts for comparison (Yao et al., 2022a;b), we can test the capability of instruction-finetuned LLM in decision making tasks in depth. The performance score is evaluated by the percentage of required attributes covered by the chosen product (from 0 to 100), and if the product meets all the requirements, that episode is labeled a success.

Because WebShop does not have API to get the screenshot of rendered websites, we focus on WebGUM with text inputs, parsed from noisy HTML in the real world.[9] We convert the actions from raw texts (e.g. `search[a long clip-in hair extension]` or `click[<item id>]`) to dictionary-like format (e.g. `{"action": "search", "ref": "a long clip-in hair extension"}` or `{"action": "click", "ref": "<item id>"}`), as we use in MiniWoB++, to improve the prediction accuracy. We finetune Flan-T5-XL with about 1K human demonstrations curated by Yao et al. (2022a)[10], using only high-score demonstrations. The score threshold is `score ≥ 50` and we have 840 episodes in total (Table 12). We construct the model input with action history, instruction, and text observation, the same as MiniWoB++ experiments. We evaluate our method with 500 user instructions in the test set.

Table 11 shows that WebGUM achieves 45.0% success, significantly outperforming not only simple baselines, such as supervised imitation learning (IL) and IL plus RL-finetuing (by more than 15%), but also recent prompt-based LLM agents, including ReAct (Yao et al., 2022b) (i.e. PaLM-540B (Chowdhery et al., 2022) with one-shot prompt and reasoning annotations), while our model only has 3 billion parameters. IL and IL plus RL-finetuning baselines use BART (Lewis et al., 2019) model for the search policy, and BERT (Devlin et al., 2019) model for the click policy. The better performance of WebGUM proves the hypothesis that the ability of multi-step reasoning in instruction-finetuned language models works as a prior for decision making problems.

| Methods | Training | Model | Modality | Score | Success Rate |
|---|---|---|---|---|---|
| Rule | – | – | Text | 45.6 | 9.6% |
| IL | SL | BART, BERT | Text(+Image) | 59.9 | 29.1% |
| IL+RL | SL+RL | BART, BERT | Text(+Image) | 62.4 | 28.7% |
| Act | In-context | PaLM-540B | Text | 62.3 | 30.1% |
| ReAct | In-context | PaLM-540B | Text | 66.6 | 40.0% |
| WebN-T5 | SL | T5-XL | Text | 61.0 | 29.8% |
| WebGUM | SL | Flan-T5-XL | Text | **67.5** | **45.0%** |
| Human | – | – | Text+Image | 82.1 | 59.6% |

Table 11: Average score and success rate on WebShop (Yao et al., 2022a) benchmark. WebGUM based on Flan-T5-XL achieves 45.0% success, outperforming most baseline approaches including ReAct, a prompted PaLM-540B with reasoning annotations. We refer Yao et al. (2022b) for the baselines.

| Threshold | # of Episodes | Score | Success Rate |
|---|---|---|---|
| `score ≥ 0` | 1026 | 67.2 | 44.4% |
| `score ≥ 50` | 840 | **67.5** | **45.0%** |
| `score = 100` | 497 | 65.3 | 44.4% |

Table 12: Average score and success rate on WebShop with different score thresholds. Because we should balance the dataset coverage and proficiency, we choose 50 as a threshold.

---

[9]WebShop just provides visual features of item pictures when the agents reach the product page. These features are extracted by ResNet-50 (He et al., 2016), rather than raw images or screenshots of the website. Some baseline agents (IL and IL+RL) incorporate such embeddings.

[10]https://github.com/princeton-nlp/WebShop/tree/master/baseline_models/data

## L    EXAMPLE EPISODES OF WEBGUM

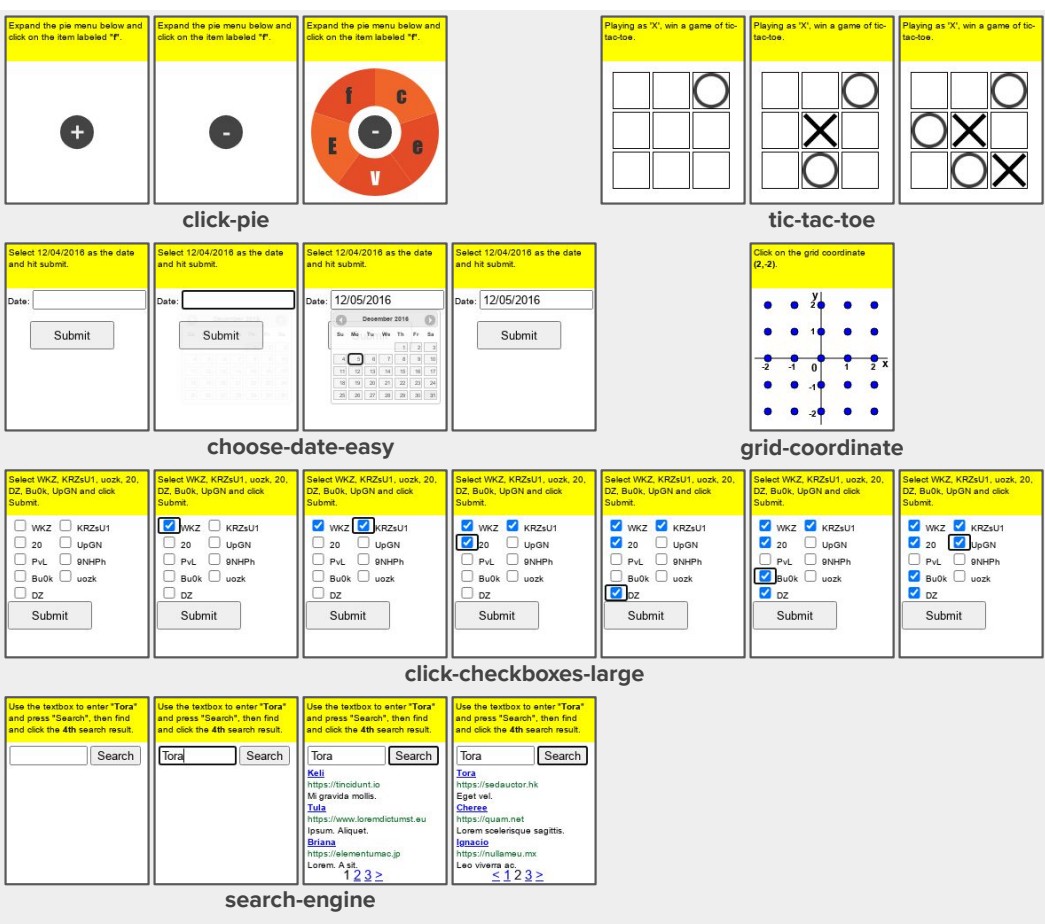

Figure 10: Example of successful episodes demonstrated by multimodal WebGUM on MiniWoB++ (Shi et al., 2017; Liu et al., 2018). The time step goes from left to right. As discussed in Section 5.1, image modality seems to be leveraged for multi-step tasks with dynamic page transitions (e.g. search-engine, choose-date-easy) or tasks that require global visual contexts (e.g. tic-tac-toe).

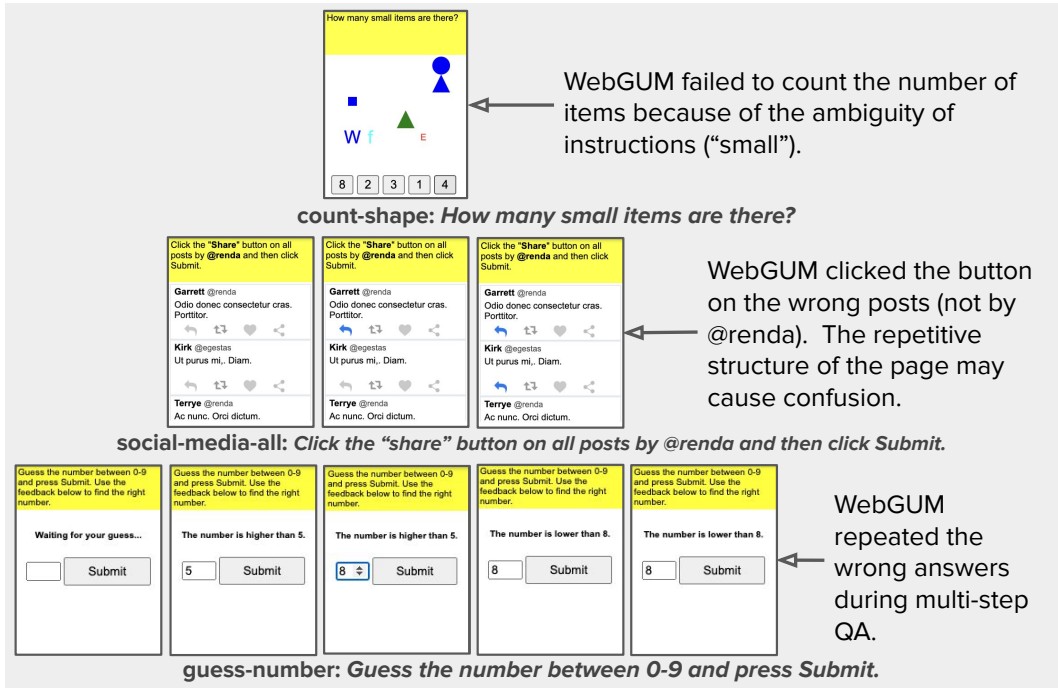

Figure 11: Example failure episodes by multimodal WebGUM on MiniWoB. The time step goes from left to right. For instance, WebGUM suffers from (1) ambiguous instruction, such as small items (count-shape), (2) confusion with the repetitive structure on the page (social-media-all), and (3) long-horizon (guess-number) tasks that may require memory.

**Instruction:** i am looking for dark denim color ethylene vinyl ultra train of size 10, 3rd generation for men, and price lower than 160.00 dollars

**Action:** search [dark denim color ethylene vinyl ultra train of size 10, 3rd generation for men]
**Observation:**
[back to search] page 1 (total results: 50) [next]

[b08b1mcfkl]
salewa ultra train 3 - men's
$104.0 to $182.99
[b07j5mc2x1]
skechers men's low-top trainers, us men
$53.39 to $140.35
[b07x5hr3tz]
bruno marc men's linen canvas stretch loafer shoes
$39.99

**Action:** click [b08b1mcfkl]
**Observation:**
[back to search] [prev]
size [7] [7.5] [8] [8.5] [9] [9.5] [10] [10.5] [11] [11.5] [12] [12.5] [13]
color [ombre blue | red orange] [raw green | black out] [dark denim | red orange]
salewa ultra train 3 - men's
price: $104.0 to $182.99
rating: n.a.
[description] [features] [reviews] [buy now]

**Action:** click [dark denim | red orange]
**Observation:** you have clicked dark denim | red orange.

**Action:** click [10]
**Observation:** you have clicked 10.

**Action:** click [buy now]
**Termination:** Your score (min 0.0, max 1.0): 1.0

---

**Instruction:** i would like a xlarge plus red camellia fleece jacket that can be machine washed, and price lower than 130.00 dollars

**Action:** search [xlarge plus red camellia fleece jacket that can be machine washed]
**Observation:**
[back to search] page 1 (total results: 50) [next]

[b09h2qsj5p]
men's thermal underwear pants usb heated warm cotton with zone 3 heating long trousers leggings bottoms
$56.77
[b09hcddf1x]
men's thermal underwear pants usb heated warm cotton long trousers leggings bottoms for winter washable baselayer
$56.77 to $56.99
[b09jgmr43j]
premium soft and smooth faux fur loose plush vest jackets fashion sexy fuzzy sleeveless outwear
$9.47 to $12.47

**Action:** click [next]
**Observation:**
[back to search] page 2 (total results: 50) [prev] [next]

[b07cqt8rbl]
king and queen write hoodie couple matching sweater pullover hooded sweatshirt jacket
$26.99 to $29.99
[b09jmvfpzg]
mens sherpa quilted lined flannel jacket plus size plaid button down fleece shirts coats thicken warm cotton outwear
$36.79 to $55.99
[b09kxfgbn2]
mankalvxy womens jacket cardigan flannel cowl neck asymmetric loose outerwear plus size xs-xxl 0-22
$20.3 to $26.85

**Action:** click [b07cqt8rbl]
**Observation:**
[back to search] [prev]
size [king-small] [queen-small] [king-medium] [queen-medium] [king-large] [queen-large] [king-xxl] [king-xlarge] [queen-xlarge] [queen-xxlarge]
king and queen write hoodie couple matching sweater pullover hooded sweatshirt jacket
price: $26.99 to $29.99
rating: n.a.
[description] [features] [reviews] [buy now]

**Action:** click [prev]
**Observation:**
[back to search] page 2 (total results: 50) [prev] [next]

[b07cqt8rbl]
king and ...
[b09jmvfpzg]
mens sherpa ...
[b09kxfgbn2]
mankalvxy womens...

**Action:** click [b09jmvfpzg]
**Observation:**
[back to search] [prev]
color [army green] [black] [coffee] [gray] [green] [red] [a-army green] [a-red]
size [small] [medium] [large] [x-large] [xx-large] [3x-large]
mens sherpa quilted lined flannel jacket plus size plaid button down fleece shirts coats thicken warm cotton outwear
price: $36.79 to $55.99
rating: n.a.
[description] [features] [reviews] [buy now]

**Action:** click [red]
**Observation:** you have clicked red.

**Action:** click [x-large]
**Observation:** you have clicked x-large.

**Action:** click [buy now]
**Termination:** Your score (min 0.0, max 1.0): 1.0

Table 13: Example of successful episodes demonstrated by WebGUM on WebShop (Yao et al., 2022a). WebGUM could search and choose the required product and choose proper options (left). In addition, WebGUM could also compare the products with browsing and backtracking (i.e. clicking "next" or "prev" buttons) during the episodes (right).

| Example 1 | Example 2 |
|---|---|
| **HTML:**
<html><div><div role="navigation"><ul><a backend_node_id="5124" title="Scores/Schedule"><text>Scores/Schedule</text></a><a backend_node_id="7499" title="GameChannel"><text>GameChannel</text></a></ul></div><div><text>HOW</text></div><div><text>68</text></div></div><ul><a backend_node_id="8365"><h3><text>Chargers' Adderley retiring from NFL at 25</text></h3></a><a backend_node_id="8641"><text>New Detroit Lions RB David Montgomery excited to join 'team that's starting something crazy'</text></a><p><text>David Montgomery, the Lions' biggest free-agent addition on offense, ran for 801 yards and five touchdowns with the Chicago Bears in 2022</text></p></div></ul></div></html> | **HTML:**
<html><body><div><label><text>Where?</text></label><input backend_node_id="12940" type="text" placeholder="Start typing or select a destination"/><button type="button"><text>×</text></button></div><section role="main"><a title="$149 & up – South Florida hotel by the beach"></a></section><ul><li backend_node_id="15241"><div><text>Near Me</text><text>Set</text></div></li><li backend_node_id="15254"><div><text>Las Vegas, NV</text></div></li><li backend_node_id="15278"><div><text>Miami, FL (Area)</text></div></li></ul></body></html> |
| **Input:**
Based on the HTML webpage above, try to complete the following task:
Task: Find the results of the most recent NFL games.
Previous actions:
`[link] NFL . -> CLICK`
What should be the next action? Please select from the following choices (If the correct action is not in the page above, please select A. 'None of the above'):

A. None of the above
B. <a backend_node_id="5124" title="Scores/Schedule"><text>Scores/Schedule</text></a>
C. <a backend_node_id="7499" title="GameChannel"><text>GameChannel</text></a>
D. 
E. <a backend_node_id="8365"><h3><text>Chargers' Adderley retiring from NFL at 25</text></h3></a>
F. <a backend_node_id="8641"><text>New Detroit Lions RB David Montgomery excited to join 'team that's starting something crazy'</text></a> | **Input:**
Based on the HTML webpage above, try to complete the following task:
Task: Find hotel deals in Las Vegas for four adults starting on May 17 and ending on May 20, and if deal is not available, set an alert for the same.
Previous actions:
`[textbox] What type of deals?  -> CLICK`
`[div] Hotels -> CLICK`
What should be the next action? Please select from the following choices (If the correct action is not in the page above, please select A. 'None of the above'):

A. None of the above
B. <input backend_node_id="12940" type="text" placeholder="Start typing or select a destination"/>
C. 
D. <li backend_node_id="15241"><div><text>Near Me</text><text>Set</text></div></li>
E. <li backend_node_id="15254"><div><text>Las Vegas, NV</text></div></li>
F. <li backend_node_id="15278"><div><text>Miami, FL (Area)</text></div></li> |
| **Prediction:** `B. CLICK` ✔ | **Prediction:** `B. TYPE las vegas` ✔ |

Table 14: Example outputs of WebGUM in Mind2Web dataset as evaluated in Section 5.5.

| Liu et al. (2018) | Ours |
|---|---|
| **Instruction:** Select xj, 9jH, KFSZqqQ, JX16, mKgO, mVVdsdH, MKJH, KLv, 8xLcf8M, YyWt5j, fS4U09Q, a130 and click Submit. | **Instruction:** Select 4yWiUvZ, Cq5, 1Lz, MlsUZU, UOIWpdw, GCM, V5qh, fk18uv8 and click Submit. |
| **HTML:**
<body ref="1"><div id="wrap" ref="2"><div id="area" ref="3"><div id="boxes-left" ref="4"><label ref="5"><input type="checkbox" id="ch0" ref="6" value="False"></input><t class="TEXT_CLASS" ref="None">KLv</t></label><label ref="7"><input type="checkbox" id="ch1" ref="8" value="False"></input><t class="TEXT_CLASS" ref="None">YyWt5j</t></label><label ref="9"><input type="checkbox" id="ch2" ref="10" value="False"></input><t class="TEXT_CLASS" ref="None">mVVdsdH</t></label><label ref="11"><input type="checkbox" id="ch3" ref="12" value="False"></input><t class="TEXT_CLASS" ref="None">9jH</t></label><label ref="13"><input type="checkbox" id="ch4" ref="14" value="False"></input><t class="TEXT_CLASS" ref="None">KFSZqqQ</t></label><label ref="15"><input type="checkbox" id="ch5" ref="16" value="False"></input><t class="TEXT_CLASS" ref="None">mKgO</t></label></div><div id="boxes-right" ref="17"><label ref="18"><input type="checkbox" id="ch6" ref="19" value="False"></input><t class="TEXT_CLASS" ref="None">JX16</t></label><label ref="20"><input type="checkbox" id="ch7" ref="21" value="False"></input><t class="TEXT_CLASS" ref="None">a130</t></label><label ref="22"><input type="checkbox" id="ch8" ref="23" value="False"></input><t class="TEXT_CLASS" ref="None">8xLcf8M</t></label><label ref="24"><input type="checkbox" id="ch9" ref="25" value="False"></input><t class="TEXT_CLASS" ref="None">xj</t></label><label ref="26"><input type="checkbox" id="ch10" ref="27" value="False"></input><t class="TEXT_CLASS" ref="None">MKJH</t></label><label ref="28"><input type="checkbox" id="ch11" ref="29" value="False"></input><t class="TEXT_CLASS" ref="None">fS4U09Q</t></label></div><button id="subbtn" class="secondary-action" ref="30">Submit</button></div></div></body> | **HTML:**
<body ref="1"><div id="wrap" ref="2"><div id="area" ref="3"><div id="boxes-left" ref="4"><label ref="5"><input type="checkbox" id="ch0" ref="6" value="False"></input><t id="None" class="TEXT_CLASS" ref="None">GCM</t></label><label ref="7"><input type="checkbox" id="ch1" ref="8" value="False"></input><t id="None" class="TEXT_CLASS" ref="None">MlsUZU</t></label><label ref="9"><input type="checkbox" id="ch2" ref="10" value="False"></input><t id="None" class="TEXT_CLASS" ref="None">fk18uv8</t></label><label ref="11"><input type="checkbox" id="ch3" ref="12" value="False"></input><t id="None" class="TEXT_CLASS" ref="None">4yWiUvZ</t></label><label ref="13"><input type="checkbox" id="ch4" ref="14" value="False"></input><t id="None" class="TEXT_CLASS" ref="None">gAVBe</t></label><label ref="15"><input type="checkbox" id="ch5" ref="16" value="False"></input><t id="None" class="TEXT_CLASS" ref="None">V5qh</t></label></div><div id="boxes-right" ref="17"><label ref="18"><input type="checkbox" id="ch6" ref="19" value="False"></input><t id="None" class="TEXT_CLASS" ref="None">1Lz</t></label><label ref="20"><input type="checkbox" id="ch7" ref="21" value="False"></input><t id="None" class="TEXT_CLASS" ref="None">UOIWpdw</t></label><label ref="22"><input type="checkbox" id="ch8" ref="23" value="False"></input><t id="None" class="TEXT_CLASS" ref="None">PDXX</t></label><label ref="24"><input type="checkbox" id="ch9" ref="25" value="False"></input><t id="None" class="TEXT_CLASS" ref="None">WVSB</t></label><label ref="26"><input type="checkbox" id="ch10" ref="27" value="False"></input><t id="None" class="TEXT_CLASS" ref="None">Cq5</t></label></div><button id="subbtn" class="secondary-action" ref="28">Submit</button></div></div></body> |
| **Actions:**
1. {action: click, ref: 25} (click checkbox `xj`)
2. {action: click, ref: 12} (click checkbox `9jH`)
3. {action: click, ref: 14} (click checkbox `KFSZqqQ`)
4. {action: click, ref: 19} (click checkbox `JX16`)
5. {action: click, ref: 27} (click checkbox `MKJH`)
6. {action: click, ref: 16} (click checkbox `mKgO`)
7. {action: click, ref: 27} (click checkbox `MKJH`)
8. {action: click, ref: 10} (click checkbox `mVVdsdH`)
9. {action: click, ref: 27} (click checkbox `MKJH`)
......
(continue) | **Actions:**
1. {action: click, ref: 12} (click checkbox `4yWiUvZ`)
2. {action: click, ref: 27} (click checkbox `Cq5`)
3. {action: click, ref: 19} (click checkbox `1Lz`)
4. {action: click, ref: 8} (click checkbox `MlsUZU`)
5. {action: click, ref: 21} (click checkbox `UOIWpdw`)
6. {action: click, ref: 6} (click checkbox `GCM`)
7. {action: click, ref: 16} (click checkbox `V5qh`)
8. {action: click, ref: 10} (click checkbox `fk18uv8`)
9. {action: click, ref: 28} (click `Submit` button) |

Table 15: Qualitative comparison between previous 12K episodes (Liu et al., 2018) (left) and our 347K episodes (right). The examples are taken from `click-checkboxes-large`. While previous work has included "hesitant" behaviors (e.g. clicking the same checkbox several times), our dataset has "shortest" behaviors. We manually annotate the action for readability (e.g. click checkbox `9jH`).