# OpenReview forum: "Multimodal Web Navigation with Instruction-Finetuned Foundation Models"
_ICLR.cc/2024/Conference — ICLR 2024 poster_

### Official Review · Reviewer_idei · 2023-10-26

**Soundness:** 3 good
**Presentation:** 3 good
**Contribution:** 2 fair
**Rating:** 5
**Confidence:** 3

**Summary:**

This paper proposes an instruction-following multimodal agent, WebGUM for web navigation tasks. The proposed method has improved upon the prior SOTA method (including humans and a GPT4-based agent) on MiniWob.

**Strengths:**

1. This paper is well written and easy to follow.
2. The proposed method achieved amazing performance on MiniWob.

**Weaknesses:**

1. The proposed method is neither impressive nor novel. Given the fact that multimodal MLLM can already complete various difficult tasks such as visual reasoning or science QA [1,3], achieving the highest score in MiniWob does not seem very surprising.
2. Since there is a collection of [opensourced MLLMs](https://github.com/BradyFU/Awesome-Multimodal-Large-Language-Models), which also combine a VIT-like vision encoder and an LLM, similar to the proposed WebGUM. The reviewer does not feel like the “showing another MLLM can do tasks like web navigation“ is significant enough. Especially when some of the prior MLLMs [1-3] are already using LLaMA as their base LM, the proposed method is still using T5.

[1] Liu, Haotian, et al. "Visual instruction tuning." arXiv preprint arXiv:2304.08485 (2023).

[2] Li, Bo, et al. "Otter: A multi-modal model with in-context instruction tuning." arXiv preprint arXiv:2305.03726 (2023).

[3] Dai, Wenliang, et al. “InstructBLIP: Towards General-purpose Vision-Language Models with Instruction Tuning”, arXiv preprint arXiv:2305.06500

**Questions:**

Have the authors tried LLaMA (or LLaMA2) for based LM? If yes, what are the results? If not, why not?

---

> ### Author Response · Authors · 2023-11-18
> **Author Response to Reviewer idei**
>
> We thank the reviewer for the careful reading and constructive feedback. We address your concerns below and please let us know if you have further questions.
>
> **> Weaknesses 1**
>
> The major difference between the recent multimodal LLMs and our WebGUM is the task they can deal with. The recent multimodal LLMs, such as LLaVA and InstructBLIP, focus on static inference tasks like visual reasoning or Science QA. In contrast, this paper tackles web navigation, an interactive decision making problem, which requires temporal understanding and planning over the multiple timesteps. Decision making for autonomy and Static VQA are orthogonal research fields that have their own specific challenges. We have clearly explained the focus of this paper in the abstract and introduction.
>
>
> **> Weaknesses 2**
>
> We have investigated the architectural choices for base language models, comparing Flan-T5 (220M, 770M, 3B, 11B) and Flan-PaLM-8B. Flan-T5 is the encoder-decoder model and Flan-PaLM is the decoder-only model. In Figure 4 (right), while Flan-T5 shows a clear scaling trend, Flan-PaLM-8B performs sub-optimally (72.8%), compared to similar scale Flan-T5 models (770M: 72.4%, 3B: 75.5%, 11B: 79.0%). **Our results imply that in web navigation, encoder-decoder architecture (T5) is better at understanding HTML than the decoder-only model (PaLM, LLaMA, etc)**, since encoder-decoder architecture can capture the hierarchical and nested structure of HTML well. Our results also suggest that advanced language models are not always the best for downstream finetuning tasks. Because both Flan-T5 and Flan-PaLM are instruction-tuned on the same Flan dataset, our comparison should be more fair than the case with LLaMA.
>
> Moreover, while recent LLMs employ larger model sizes (LLaMA-7B + ViT-L14 for LLaVA, LLaMA-7B/13B + ViT-G14 for InstructBLIP), WebGUM only has 3 billion parameters (Flan-T5-XL + ViT-B16). Such parameter efficiency is a desirable property for autonomous agent research because it ensures on-premise deployment and low latency.
>
> Lastly, in the revised paper we have included the works you suggested (LLaVA, Otter, InstructBLIP) in the extended related work (Appendix B).

---

### Official Review · Reviewer_cMQF · 2023-10-28

**Soundness:** 4 excellent
**Presentation:** 4 excellent
**Contribution:** 4 excellent
**Rating:** 8
**Confidence:** 4

**Summary:**

This paper studied data-driven offline training for web agents with VLMs. The proposed WebGUM observes both webpage screenshots and HTML pages and outputs web navigation actions, such as click and type. The authors collected 347K high-quality demonstrations using their trained model,

**Strengths:**

Novelty: The proposed WebGUM agent exhibits a novel combination of HTML and image modalities to tackle the challenges in web navigation.
Performance: The empirical results are compelling, with the model showing substantial improvements on the MiniWoB and WebShop benchmarks.
Resource Contribution: The authors have collected and made available a significant corpus of high-quality demonstrations, which is 38 times larger than previous datasets. This contribution is likely to be valuable for the broader research community working on similar problems.
Clarity and Structure: The paper is well written with clear explanations of the methodology and discussions surrounding the results. The inclusion of comparisons with existing SoTA methods provides a good understanding of the performance gains achieved.

**Weaknesses:**

- Generalization: It’s not clear how well the proposed method generalizes to a broader range of web navigation tasks outside the tested benchmarks, especially HTML that are longer than context length. More discussions or evaluations on the generalizability could strengthen the paper. "Because the context length is insufficient for raw HTML, we preprocess context HTML by extracting a snippet that includes the answers in advance."
- Figure 1 and 5 are blurry.

**Questions:**

- failure cases analysis: include failures cases on datasets like minwob, on which the success rate is high.
- include demos on mind2web and webshop.

---

> ### Author Response · Authors · 2023-11-18
> **Author Response to Reviewer cMQF (1/2)**
>
> We thank the reviewer for the thoughtful review and comments. Please let us know any remaining questions or concerns if you have.
>
>
> **> Weaknesses 1**
>
> As described in Appendix C, our WebGUM accepts 4096 tokens as input, which might be the soft upper limit for dense attention models considering the computational costs. As the reviewer pointed out, in the evaluation of WebSRC (Appendix E), we have preprocessed HTML to shorten it, because in some train/test sets, HTML has over 10K tokens. For web navigation tasks with much longer HTML, we could incorporate (1) ranking language models [1] to prioritize HTML elements, (2) extractive summarization of HTML [2], or (3) long-context pre-trained language models with sparse attention [3], in the pipeline. In fact, we used the cached ranking results, released by the author, for Mind2Web, and we demonstrated the applicability of such a pipelined approach by achieving the best performance among the baselines.
>
>
>
> **> Weaknesses 2**
>
> Thank you for pointing this out. We updated Figure 1 and 5 with high-resolution images in the revised paper.

---

> ### Author Response · Authors · 2023-11-18
> **Author Response to Reviewer cMQF (2/2)**
>
> **> Questions 1**
>
> We have included a failure example of WebGUM in MiniWoB in Appendix L. In addition to that, we increased the number of examples in the revised paper. Please take a look at Figure 11 (Appendix L). For instance, WebGUM suffers from (1) ambiguous instruction, such as small items (`count-shape`), (2) confusion with the repetitive structure on the page (`social-media-all`), and (3) long-horizon (`guess-number`) tasks that may require memory.
>
>
> **> Questions 2**
>
> We have included the successful episodes on WebShop in Table 13. In the revised paper, we added the example response on Mind2Web in Table 14 (Appendix L).
>
>
> ```
> [1] Deng et al., (2023) Mind2Web: Towards a Generalist Agent for the Web (https://arxiv.org/abs/2306.06070)
>
> [2] Gur et al., (2023) A Real-World WebAgent with Planning, Long Context Understanding, and Program Synthesis (https://arxiv.org/abs/2307.12856)
>
> [3] Guo et al., (2021) LongT5: Efficient Text-To-Text Transformer for Long Sequences (https://arxiv.org/abs/2112.07916)
> ```

---

### Official Review · Reviewer_a21F · 2023-11-11

**Soundness:** 3 good
**Presentation:** 3 good
**Contribution:** 2 fair
**Rating:** 5
**Confidence:** 4

**Summary:**

This paper studied web navigation with instruction-finetuning multi-modal models. Specifically, they created a large-scale datasets by generating data with pretrained LLMs. They then train a vision-language model with flan-t5 as the base language model and ViT as the vision encoder. The model will take html code and image screenshot as inputs and take navigation actions such as click and type.

**Strengths:**

1. The performance of their model is good. It outperforms previous methods under different settings.
2. Writing is clear.
3. They performed detailed analysis such as dataset and model size scaling.

**Weaknesses:**

1. The technical contribution is very limited. The takeaway is to do supervised training on a large-scale model-generated dataset. It feels like knowledge distillation of a combination of model outputs (as described in 4.3 they used various LLMs to generate such data).
2. The dataset creation process and quality is not clear.

**Questions:**

1. How is the generated dataset high-quality? Is there a human study to prove this? As described in section 4.3, my understanding is this LLM-generated dataset can be noisy.
2. It is also not very clear about the data generation process. What does it mean by "rollout a LLM policy with 100 episodes per task"? What are the inputs and outputs here?
3. As mentioned in section.5, the method is evaluated on MiniWoB++ with 100 evaluation episodes per task. Does the reported numbers from previous work also use the same eval set?
4. Could you explain more on why your model would outperform human? It is hard to imagine a small model can do better than human on general web navigation. Is it due to overfitting to a specific dataset (Table.1)?

---

> ### Author Response · Authors · 2023-11-18
> **Author Response to Reviewer a21F (1/2)**
>
> We sincerely appreciate your thorough examination of our paper and your insightful feedback. We have carefully considered your concerns and offer the following clarifications and additional information. Please let us know if there are remaining questions or unclear points.
>
>
> **> Weaknesses 1**
>
> We have proposed a simple yet effective recipe to build multimodal web navigation agents, leveraging the prior knowledge of the web environment from instruction-finetuned language models and visual encoders. We believe that our paper has the following important technical contributions:
>
> (1) We have introduced temporal and local visual tokens to perceive visual information better (Figure 2 and Section 5.1). Given the interactive nature of web navigation on computers, it is essential to consider both temporal and local relationships on the screen. Our experimental evaluation in Figure 4 (left) indicates that employing both temporal and local perception tokens is most critical for the performance, while the effects of various pre-trained ViT with different datasets or self-supervised objectives are marginal.
>
> (2) We have investigated the architectural choices for base language models, comparing Flan-T5 (220M, 770M, 3B, 11B) and Flan-PaLM-8B. Flan-T5 is the encoder-decoder model and Flan-PaLM is the decoder-only model. In Figure 4 (right), while Flan-T5 shows a clear scaling trend, Flan-PaLM-8B performs sub-optimally (72.8%), compared to similar scale Flan-T5 models (770M: 72.4%, 3B: 75.5%, 11B: 79.0%). Our results imply that in web navigation, encoder-decoder architecture is better at understanding HTML than the decoder-only model, since encoder-decoder architecture can capture the hierarchical and nested structure of HTML well.
>
> (3) We have shown a good estimation of the necessary amount of training dataset for multimodal web navigation. Previously, the community only had 12K, HTML-only dataset, and CC-Net implied that 2.4 million episodes and billions of frame online interactions would be needed to realize human-level performance. Our human-level performance with 401K demonstrations, and the newly introduced 347K multimodal dataset should encourage the community to develop capable on-premise multimodal agents via safe offline training.
>
>
> **> Weaknesses 2 & Questions 1**
>
> As described in Appendix F, we run LLM agents with 10,000 episodes per task and only keep successful trajectories to maintain the quality of the dataset. Such reward filtering is a common practice in web navigation as done in Humphreys et al., (2022) [2]. Because our dataset only consists of successful episodes, they do not have noisy data (i.e. failure episodes). We also added qualitative analysis in Table 15 (Appendix L) comparing the episodes from our 347K dataset and the previous 12K dataset. This shows that while previous work has included "hesitant" behaviors (e.g. clicking the same checkbox several times), our dataset has "shortest" behaviors. The quality of our dataset is appropriately controlled.
>
>
> **> Questions 2**
>
> "rollout" means "deploy" here. We deployed WebN-T5 (i.e. LLM policy) on MiniWoB environments, and WebN-T5 was asked to solve 100 episodes per task for all 56 tasks. The input and output are the same as our WebGUM; taking HTML, instruction, and action history as inputs, and predicting dictionary-format action (as described in Figure 2).
>
>
>
> **> Questions 3**
>
> Yes. Following Gur et al., (2022) [1] and Humphreys et al., (2022) [2], we evaluate WebGUM with 100 episodes per task. While other LLM baselines (RCI, AdaPlanner, Synapse) are evaluated with only 50 episodes, our evaluation with more episodes could reduce the variance during the evaluation.

---

> ### Author Response · Authors · 2023-11-18
> **Author Response to Reviewer a21F (2/2)**
>
> **> Questions 4**
>
> We’d like to note that, in web navigation literature, the number of parameters does not always correlate to the performance. For instance, CC-Net [2], trained through supervised learning and reinforcement learning, only has small-scale architectures compared to LLMs (transformer with 8 layers, 8 heads, and a 512-dimensional embedding, a dual-layer LSTM with 512
> hidden units per layer, and 4 blocks ResNet). However, CC-Net can exhibit human-level web navigation performance (93.5%). Among the human-level web navigation methods (CC-Net, WebGUM, RCI/Synapse (GPT-3.5/4 with prompting)), there are trade-offs between model size, training cost/risk, and inference cost. Our method provides promising solutions among those by (1) mitigating training costs and risks through offline supervised finetuning and (2) achieving on-premise agents with low inference costs.
>
> | | CC-Net | WebGUM | RCI / Synapse |
> |--|--|--|--|
> | **Model Size** | small | up to 3B parameters | API-based private LLM |
> | **Training Cost/Risk** | High (RL) | Low (SL) | -- (in-context)|
> | **Inference Cost** | Low | Low | High |
>
> In addition, we have conducted an out-of-distribution evaluation in Section 5.3 (Figure 5, and 6), and a transfer learning evaluation in Section 5.5 (Table 3). WebGUM performs the best among competitive baselines. In particular, WebGUM achieved 78.5% success rate for unseen combinational tasks, outperforming Synapse (73.8%) a SoTA web navigation agent in Table 1. In real-world action prediction tasks from the Mind2Web dataset, WebGUM exhibits strong transfer achieving the best performance in all the metrics. These results support that WebGUM is generalizable enough to other web navigation tasks beyond overfitting.
>
>
>
> ```
> [1] Gur et al., (2022) Understanding HTML with Large Language Models (https://arxiv.org/abs/2210.03945)
>
> [2] Humphreys et al., (2022) A data-driven approach for learning to control computers (https://arxiv.org/abs/2202.08137)
> ```

---

### Author Response · Authors · 2023-11-18
**Summary of Revision in Author Response**

We would like to appreciate the thoughtful comments from all the reviewers. We revised the manuscript based on your constructive feedback and suggestions (**highlighted in purple**). The key changes are summarized below:


- Replace Figure 1 and Figure 5 with high-resolution images.
- Update failure examples on MiniWoB in Figure 11 (Appendix L).
- Add example outputs for Mind2Web in Table 14 (Appendix L).
- Add a qualitative comparison between the previous 12K dataset and ours in Table 15 (Appendix L).


Lastly, we’d like to highlight our contribution to address the reviewer’s concerns:

**Focus of This Paper (Reviewer idei):**

This paper tackles web navigation, an interactive decision making problem requiring temporal understanding and planning across multiple timesteps. On the other hand, recent multimodal LLMs, such as LLaVA and InstructBLIP, focus on static inference tasks like visual reasoning or Science QA. Decision making for autonomy and Static VQA are orthogonal research fields that have their specific challenges. We have clearly explained the focus of this paper in the abstract and introduction.


**Dataset Quality (Reviewer a21F):**

As detailed in Appendix F, we run LLM agents with 10,000 episodes per task and only keep successful trajectories to maintain the dataset's quality. Such reward filtering is a common practice in web navigation as done in Humphreys et al., (2022) [2]. **Because our dataset only consists of successful episodes, they do not have noisy data (i.e. failure episodes).** Therefore, the quality of our dataset is appropriately controlled. In addition, we added a qualitative analysis in Table 15 (Appendix L) comparing the episodes from our 347K dataset and the previous 12K dataset. This shows that while previous work has included "hesitant" behaviors (e.g. clicking the same checkbox several times), our dataset has "shortest" behaviors.


**Encoder-Decoder Language Models for Web Navigation (Reviewer a21F & idei):**

Our experimental evaluation reveals that Encoder-Decoder language models, specifically Flan-T5, outperform Decoder-only models, like Flan-PaLM, in the context of web navigation. The hierarchical and nested structure of HTML is better captured by the encoder-decoder architecture. Figure 4 (right) illustrates the suboptimal performance of Flan-PaLM-8B (72.8%) compared to similar-scale Flan-T5 models (770M: 72.4%, 3B: 75.5%, 11B: 79.0%). Importantly, our results challenge the assumption that more advanced language models are universally superior for downstream fine-tuning tasks. Notably, the comparison between Flan-T5 and Flan-PaLM is particularly fair, given that both models are instruction-tuned on the same Flan dataset, a contrast with the situation with LLaMA.


**On-premise Agents with Human-Level Performance (Reviewer a21F & idei):**

Among the human-level web navigation methods, including CC-Net, WebGUM, and RCI/Synapse (GPT-3.5/4 with prompting), there are trade-offs between model size, training cost/risk, and inference cost. Our method provides promising solutions among those by (1) mitigating training costs and risks through offline supervised finetuning and (2) achieving on-premise agents with low inference costs.

We hope our revision and response to each reviewer address your concerns. Please feel free to raise any remaining questions or concerns you may have.

---

> ### Author Response · Authors · 2023-11-22
>
> Dear Reviewers,
>
> We would appreciate it if they could check our updates and feel free to raise further questions if you have. We are happy to clarify them. Thank you so much for your time!
>
> Sincerely,
>
> Authors

---

### Meta-Review · Area_Chair_P2hV · 2023-12-03

**Metareview:**

(a) The paper addresses the problem of web navigation, which involves making interactive decisions based on temporal information and planning over multiple steps. The paper proposes a method that takes as inputs webpage screenshots, action history, instruction, and HTML code, and produces web navigation actions as outputs. The paper claims that the method enhances the agent’s abilities to perceive multimodal information, understand HTML code, and reason across multiple steps.

(b) The advantage of this article is that it solves the task of web navigation with temporal and local perception on a large corpus of demonstrations combining the understanding capabilities of multiple models. The experimental data shows that this method does bring certain performance improvements.

(c) The shortcoming of this article is that its technical contribution is relatively limited. The fairness of comparisons between models is not well explained in the article. Moreover, the generation of the data set is not clearly described, and the generated data is not quantitatively evaluated and analyzed.

**Justification For Why Not Higher Score:**

I didn't give it a higher score because its technical contribution is indeed limited. And there is confusion in the article about the unclear construction of the data set raised by the reviewer.

**Justification For Why Not Lower Score:**

I didn't give it a lower score because it solves the task of automatic web navigation and presents a method that can be proven effective.

---

### Decision · Program_Chairs · 2024-01-16

Accept (poster)